**communications** engineering

# A multipath error cancellation method based on antenna jitter
Jiyang Liu [1,2], Feixue Wang [1,2,3] ✉, Xiaomei Tang[1,2,3], Sixin Wang [1,2,3] ✉ & Muzi Yuan [1,2,3] ✉

Global Navigation Satellite System signals are often affected by multipath errors, which impact the accuracy of positioning measurements. Traditional methods frequently fail to effectively mitigate multipath errors across different environments, primarily due to their inherent sensitivity to varying conditions. Here, we propose a multipath error cancellation method that utilizes antenna jitter, which mitigates multipath errors by rapidly changing the relative phases of direct and multipath signals without requiring changes to the receiver structure. The model that combines theoretical analysis with experimental verification is conducted to identify the minimum jitter amplitude required for effective error reduction in straight-line jitter scenarios. Moreover, extensive satellite data collection and verification were performed in Changsha, China, from December 2023 to August 2024. The results indicate that the proposed method enhances robustness and applicability across various environments compared to traditional approaches. Notably, it enables a vehicle-mounted antenna, priced at just a few dollars, to achieve positioning accuracy comparable to that of high-precision antennas costing thousands of dollars, making advanced positioning technology more accessible.

Global Navigation Satellite System (GNSS) is widely used in spatial and temporal positioning, mobile connectivity, unmanned driving, and other fields, whose rapid development also puts forward higher requirements for positioning accuracy. However, multipath error, intricately tied to the observation environment and challenging to anticipate, plays a dominant role in degrading the accuracy performance of GNSS positioning within urban canyons[1,2]. Unlike other error components such as those stemming from satellite and receiver clocks, ionospheric and tropospheric propagation, and satellite orbits, multipath error remains intractable to the differential approach[3].

Various strategies have been devised to address multipath errors, with conventional approaches typically falling into three primary classifications as outlined: antenna-based technology, algorithms for baseband processing, and post-processing methods[4,5]. In particular, the antenna-based technology forces the antenna to receive the indirect signals as little as possible. Although strategically positioning a GNSS antenna in an optimal location is the most effective method for mitigating multipath effects, achieving such ideal conditions is often impractical within the confines of urban canyons[6]. Another effective strategy to mitigate multipath interference involves the deployment of meticulously engineered antennas, encompassing both signal antennas and antenna arrays. The single antenna, such as dual-polarization antennas[7] and choke ring antennas[8], can suppress the short-delay multipath. The antenna array[9,10] requires prior knowledge of the direct

signal direction and the assistance of inertial navigation. Although these well-designed antennas mitigate multipath interference effectively, they are expensive and bulky[11].

Moreover, baseband processing algorithms are used to mitigate multipath errors through signal processing links, which can be primarily categorized into non-parametric and parametric methods[12]. On one side, the methods based on the nonparametric estimation mainly include the narrow-spacing correlation[13], the High-Resolution Correlator[14], and the strobe correlator[15]. The narrow-spacing algorithm limits multipath interference but is constrained by practical bandwidth limits, becoming ineffective when errors are below the channel's reciprocal bandwidth. Both the High-Resolution Correlator and strobe correlator variants struggle with short-delay multipath. Besides, the nonparametric estimation method has a simple structure, but its mitigation capability is not as good as the parametric estimation method[15].

On the other side, the parametric estimation method, often employing a maximum likelihood algorithm, is integral to the development of correlator array structures. Techniques like the multipath estimated delay locked loop and its enhancements[16], such as multipath mitigation technology[17], the coupled amplitude delay lock loop[18], and enhanced multipath estimated delay locked loop[19], are the main applications of parametric estimation. These methods estimate and mitigate multipath through recursive processing. However, the accuracy of such techniques is ultimately limited by

[1]College of Electronic Science and Technology, National University of Defense Technology, Changsha, China. [2]National Key Laboratory for Positioning, Navigation and Timing Technology, Changsha, China. [3]These authors contributed equally: Feixue Wang, Xiaomei Tang, Sixin Wang, Muzi Yuan.
✉e-mail: fxwang@nudt.edu.cn; wsx@nudt.edu.cn; ymz@nudt.edu.cn

correlator spacing, and their computational complexity is high, which may limit their application in real-time GNSS receivers[16].

In addition, various postprocessing methods have been developed to mitigate multipath errors. Traditional post-processing methods include the Gaussian–Newton method[20], multifrequency combination processing[21], and precise point positioning[22], which are primarily based on physical or empirical models. Recently, some interesting postprocessing methods that utilize external information have emerged, such as the terrain network-based assistance method[23], stochastic state estimation[24], and sparsity-promoting regularization[25]. Nevertheless, the additional external assistance of static, long-term observation data is required for the mentioned methods[26,27].

In summary, despite the effectiveness of traditional algorithms, they do come with notable application limitations[27,28]. Multipath errors associated with Geostationary Earth Orbit (GEO) satellites are more severe than those of Inclined Geosynchronous Orbit (IGSO) and Medium Earth Orbit (MEO) satellites, due to the relatively fixed position of GEO satellites and GNSS receivers[29]. Furthermore, researchers have identified a persistent deviation caused by multipath signals when analyzing the pseudorange observations of GEO satellites. This phenomenon, known as the "Standing Multipath", cannot be mitigated through traditional methods[30].

Fortunately, there is evidence that the dynamic relationship between GNSS receivers and satellites is crucial in multipath error. In view of this, a technique for mitigating multipath errors through antenna jitter was introduced[31]. Particularly, this method can be seamlessly implemented across various GNSS receivers without structural modifications and shows excellent adaptability to diverse environmental conditions.

Previous investigations into the cancellation of multipath errors through antenna motion have encompassed a spectrum of methodologies. The relationship between antenna motion and multipath error was initially studied by Brekel and Nee[31]. However, they did not propose or evaluate a specific multipath reduction method using an actual GNSS receiver. Subsequently, Ertan et al. utilized known antenna motions for multipath estimated delay locked loop. Nevertheless, this approach could not be implemented with existing GNSS receivers[32,33]. Moreover, a technique for detecting multipath sources using array antennas has been prestnted. This method enables the estimation of signal arrival directions but is complex and cost-prohibitive, so it is unsuitable for large-scale commercial deployment[34,35]. A recent development introduced a method to identify GNSS antenna multipath parameters based on the SNR of received signals[36]. They extracted the oscillatory motion characteristics of the antenna from the multipath signals. Furthermore, pioneered the use of rotating antennas for mitigating multipath errors, achieving notable results with a 25 cm rotating radius[37]. However, the existing studies lack an analysis of the general influencing factors of antenna jitter and the effect of the deviation of the direct signal. As a result, the practicality and safety of the research results are greatly reduced.

To address these research gaps, this paper unifies all modes of antenna jitter into linear jitter and quantitatively analyses the effects of jitter angle and amplitude. In addition, the deviation of the direct signal due to jitter has also been considered, which greatly enhances the practicality of this technique. In summary, the contributions of this thesis can be summarized as follows:

1. In this paper, all jitter methodologies have been mapped onto a linear jitter framework, substantially reducing jitter amplitude. This paper consolidates various jitter techniques into a linear model for analysis, examining the effects of jitter angle and amplitude on the suppression of multipath errors. Specifically, we identify the minimal jitter amplitude necessary for optimal multipath error cancellation with a single reflecting surface. Then, this minimum jitter amplitude is generalized to scenarios featuring multiple reflecting surfaces, offering a theoretical benchmark for jitter method selection.

2. The efficacy of the optimal amplitude for antenna jitter was rigorously validated. Empirical data from experiments substantiate that a straight-line jitter of ±6.2 cm effectively mitigates multipath errors. Moreover,

the practical utility of employing a straight-line jitter approach lies in its spatial efficiency, conserving an obvious amount of space that is advantageous for applications with constrained dimensions.

3. To ascertain the applicability of antenna jitter in various settings, a series of cancellation effect tests have been conducted across diverse environmental conditions. The experimental results demonstrate that the introduction of antenna jitter obviously diminishes the correlation between multipath errors and environmental factors. Consequently, this approach not only enhances the precision of satellite navigation positioning but also has the potential to reduce antenna-related expenses. The necessity for choke ring antennas may be obviated, leading to a more cost-effective solution in eliminating multipath error.

## Results

We conducted two experiments in different scenarios to evaluate the proposed method. Testing in different scenarios allows for verification of the effect of jitter direction on randomized multipath error and the environmental adaptability of antenna jitter. This chapter focuses on testing the antenna jitter performance based on the BeiDou signal in both an open platform and an artificially complex multipath platform. Furthermore, to empirically highlight the merits of the antenna jitter technique, we have implemented our method on an easy vehicular antenna and conducted a comparative analysis with a high-precision monitoring antenna.

### Open platform

The open platform multipath environment is favorable as it allows for calibrating the main reflective surface. Moreover, azimuth notation accurately depicts the antenna jitter direction and satellite position in this experiment, where 0° indicates the north direction and clockwise increments after that. The antenna undergoes jitter along a straight line facilitated by a motorized slide. To further clarify, Fig. 1 provides a schematic representation of the measured environment, outlining the spatial relationship between the multipath source from the GEO satellite and the direction of the antenna jitter.

In Fig. 1, Beidou No. 1 and No. 59 satellites have azimuths of 127° and 133°, respectively, where the main reflective surface is the wall in the 300° direction. According to theoretical analysis, the optimal jittering mode for the antenna involves moving it in a straight line along the normal of the reflecting surface. Therefore, the best antenna jittering direction for satellites No. 1 and No. 59 is 120°. Additionally, we arrange jittering in the 210° direction, parallel to the reflecting surface, for comparison with the optimal jittering direction. The specific experimental parameters are listed in

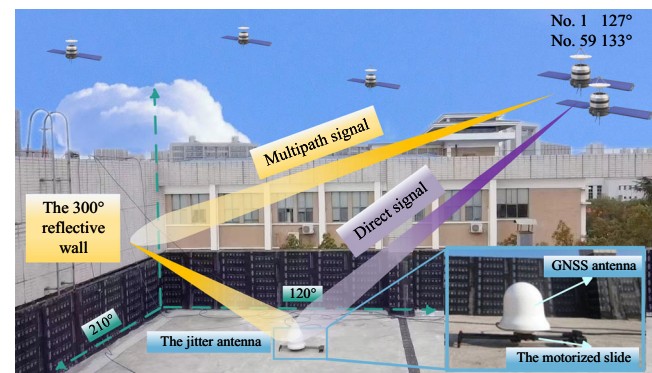

**Fig. 1 | Measured scene in open platform.** The figure provides a schematic representation of the measured environment, outlining the spatial relationship between the multipath source from the GEO satellite and the direction of the antenna jitter. Beidou No. 1 and No. 59 satellites have azimuths of 127° and 133°, respectively, where the main reflective surface is the wall in the 300° direction. The best antenna jittering direction for satellites No. 1 and No. 59 is 120°. Additionally, we arrange jittering in the 210° direction, parallel to the reflecting surface, for comparison with the optimal jittering direction.

**Table 1 | Parameters of open field experimental platform**

| Parameters | Values |
|---|---|
| Site | Changsha, China |
| Antenna | Monitoring antenna |
| GNSS receiver | Monitoring receiver |
| Positioning mode | B1I |
| Jitter angle | 120°/210° |
| Jitter amplitude | ±6.2 cm |
| Angle α (reflective surface) | 90° |
| Angle β (No.1/No.59) | 41.4°/ 40.3° |
| Motorized slide speed | 5 cm/s |

**Table 2 | Result of single-point positioning Root Mean Square (RMS) and Maximum error in the open platform**

| Antenna type | RMS error (m) | Maximum error (m) |
|---|---|---|
| Static | 2.45 | 18.16 |
| 120° | 1.51 | 15.49 |
| 210° | 1.58 | 14.64 |

Table 1, where the product information for the antenna and receiver can be found in Supplementary Note 1.

The theoretical minimum jitter amplitude can be derived through Eq. (1) based on Table 2 and Theorem 1 in method chapter, where the carrier frequency $f_c = 1561.098\text{MHz}$

$$r_M = \frac{c}{4f_c|\sin(\alpha+\beta)\cos\psi_n|} \approx 6.2\text{cm} \tag{1}$$

Furthermore, the procedure for testing the cancellation of multipath errors based on antenna jitter has the following steps: The first step is to measure precise raw data. Subsequently, the two-frequency formula is applied to extract the multipath error. It should be noted that the receiver used in this experiment outputs data once per second, which has been smoothed.

**Procedure for testing the cancellation of multipath errors based on antenna jitter.**

1. Set up the antenna and slide rail. Place the antenna and slide 5 m from the wall in the 300° direction and 5 m from the low wall in the 30° direction.

2. Raw data were collected when the remains were static, jittered in the 120° direction, and jittered in the 210° direction.

3. Pseudorange ($\rho_i$), wavelength ($\lambda_i$), carrier frequency ($f_i$), and carrier phase ($\theta_i$) were solved for different frequency points.

4. Calculate the multipath error for different frequency points.
$M_{\rho 1} = \rho_1 - \frac{f_1^2+f_2^2}{f_1^2-f_2^2}\lambda_1\phi_1 + \frac{2f_2^2}{f_1^2-f_2^2}\lambda_2\phi_2$

$M_{\rho 2} = \rho_2 - \frac{2f_2^2}{f_1^2-f_2^2}\lambda_1\phi_1 + \frac{f_1^2+f_2^2}{f_1^2-f_2^2}\lambda_2\phi_2$

To quantitatively compare the effect of the proposed method, we selected two statistics: MS error and range (max-min) error, where Fig. 2 demonstrates the extraction of multipath error for select satellites under different conditions, including static antenna, antenna jitter at 120° direction, and jitter at 210° direction. Additionally, a comparative plot of multipath error statistics for all observed satellites is presented in Fig. 3.

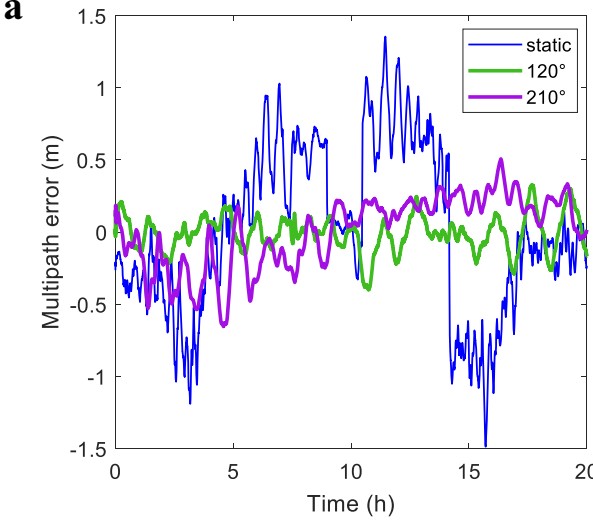

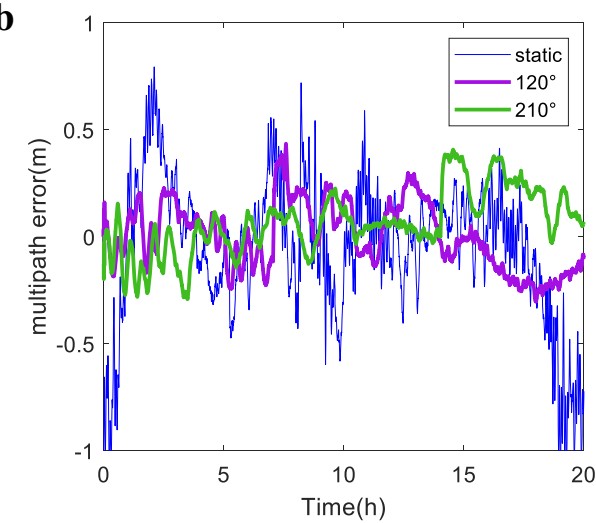

**Fig. 2 | Multipath error in the open platform. a** NO. 59, and **b** NO.1. The figure demonstrates the extraction of multipath error for select satellites, where (**a**) and (**b**) illustrate satellite 59 and 1, respectively. There was an obvious effect in eliminating the original constant error of GEO satellites with antenna jitter. The code multipath error of the static antenna is slowly changed in the figure. In contrast, the multipath error of the jitter antenna is smaller than that of a static antenna. Specifically, MS error for the B1 carrier frequency for the No.1 satellite, decreased from 0.33 m to 0.15 m post antenna jitter. Furthermore, the range error is reduced from 2.08 m to 0.76 m. This indicates that the multipath error can be suppressed by the jittering antenna.

As can be seen from Figs. 2 and 3, antenna jitter has an obvious effect on suppressing multipath errors, which was evidenced by a substantial reduction in the mean square (MS) error and range error values. In particular, the following conclusions can be drawn:

1. There was an obvious effect in eliminating the original constant error of GEO satellites with antenna jitter, where the specific data of before and after jitter are shown in Fig. 2. It is that the code multipath error of the static antenna is slowly changed in Fig. 2. In contrast, the multipath error of the jitter antenna is smaller than that of a static antenna. Specifically, MS error for the B1 carrier frequency for the No.1 satellite, decreased from 0.33 m to 0.15 m post antenna jitter. Furthermore, the range error is reduced from 2.08 m to 0.76 m. This indicates that the multipath error can be suppressed by the jitter antenna.

2. The measured optimal jitter mode aligns with theoretical expectations. For the B1 carrier frequency, the MS error of jitter in the 120° and 210° antenna of the No.1 satellite is reduced to 45.45% and 48.48% of the

## a

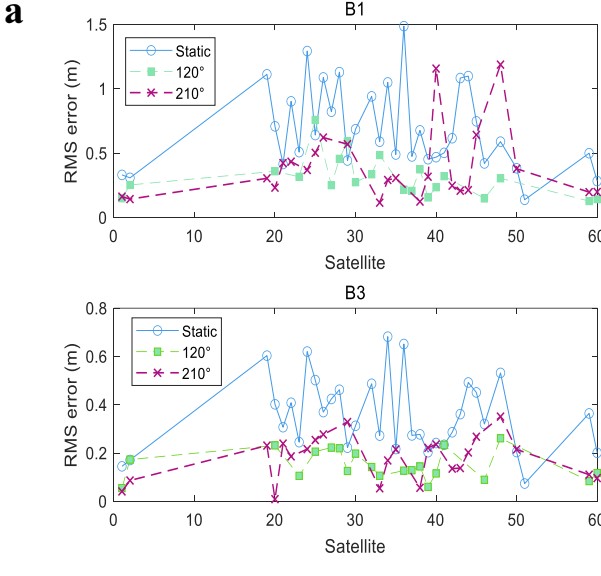

## b

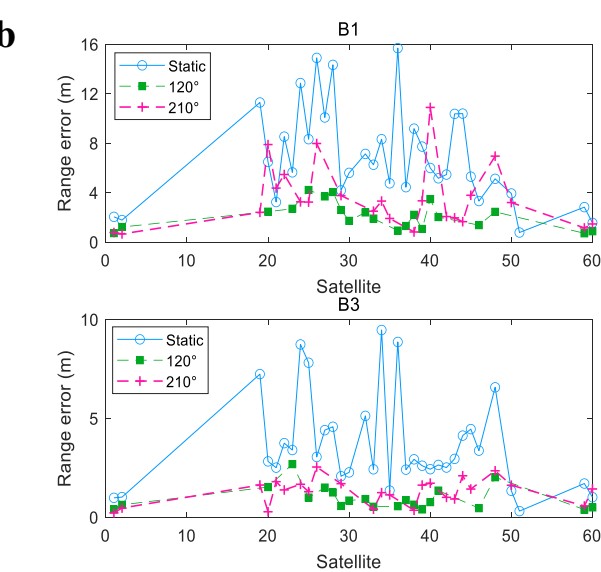

**Fig. 3 | Statistical data of multipath error in open platform. a** Mean square error, and **b** Range error. A comparative plot of multipath error statistics for all observed satellites is presented in the figure, where (**a**) and (**b**) illustrate the mean square (MS) error and Range error, respectively. The measured optimal jitter mode aligns with theoretical expectations. For the B1 carrier frequency, the MS error of jitter in the 120° and 210° antenna of the No.1 satellite is reduced to 45.45% and 48.48% of the static antenna, respectively. Similarly, for the No.59 satellite, the MS error of the 120° and 210° antenna reduced to 23.12% and 32.52% of the static antenna, respectively. This analysis demonstrates that jitter elimination is more effective along the axis perpendicular to the vertical reflective surface than in the parallel direction, thereby validating the optimal jitter approach.

static antenna, respectively. Similarly, for the No.59 satellite, the MS error of the 120° and 210° antenna reduced to 23.12% and 32.52% of the static antenna, respectively. This analysis demonstrates that jitter elimination is more effective along the axis perpendicular to the vertical reflective surface than in the parallel direction, thereby validating the optimal jitter approach. However, the experimental setup's open platform does not allow for the exclusive presence of a single reflective surface to be fully controlled. Consequently, the elimination effect is still observed when jittering in the direction parallel to the reflective surface, which is consistent with the simulation outcomes.

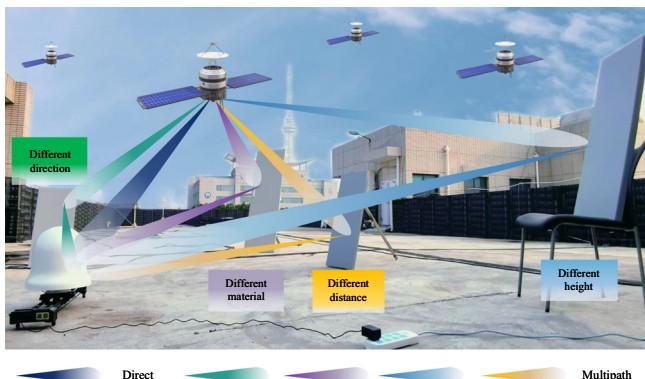

**Fig. 4 | Placement of the artificially complex platform.** The figure displays the experimental placement of an artificially complex platform. To further validate the adaptability of antenna jitter, a complex multipath environment created by human intervention is employed for testing. Building upon the initial environment setup, iron plates with specular reflection properties are strategically positioned around the antenna, including iron plates of different material, direction, distance, and height.

### Table 3 | Result of multipath multipath Mean Square (MS) and range error in the complex platform

| Antenna type | PRN = 1 | | PRN = 59 | |
|---|---|---|---|---|
| | MS error (m) | Range error (m) | MS error (m) | Range error (m) |
| Static | 0.53 | 3.12 | 1.20 | 11.54 |
| 120° | 0.14 | 0.86 | 0.11 | 0.52 |
| 210° | 0.21 | 3.05 | 0.33 | 2.16 |

## Artificially complex platform

To further validate the adaptability of antenna jitter in varied environments, a complex multipath environment created by human intervention is employed for testing. Building upon the initial environment setup, iron plates with specular reflection properties are strategically positioned around the antenna. Specifically, Fig. 4 displays the experimental placement of an artificially complex platform. It is apparent from Supplementary Fig. 1 that even in the complex platform, antenna jitter can still play a role in suppressing multipath errors. Specifically, the following conclusions can be drawn:

1. A notable observation from Supplementary Fig. 1 is that the antenna jitter exhibits a more pronounced multipath error reduction in complex platforms than open platforms. As the complexity of the experimental environment escalates, there is an obvious increase in the MS error for static antennas. In contrast, the MS error for jitter antennas (120° or 210°) displays a far less pronounced change. Furthermore, Supplementary Fig. 1 illustrates a diminished difference in the jitter direction between 120° and 210° (represented by the red and green lines). This suggests that in complex environments, the sensitivity to the direction of jitter is lessened, indicating that antenna jittering is more adaptable to varying environmental conditions. To enable a direct comparison with the open platform, Table 3 presents a comprehensive statistical analysis for satellites No. 1 and No. 59.

2. Antenna jitter has an additional eliminating effect on the MEO satellites. Although the multipath error of MEO satellites is inherently dynamic, it requires tens of minutes of sampling to eliminate by averaging. What stands out is that antenna jitter can reduce that time to seconds. Specifically, the MS error of MEO satellites can be decreased from 0.89 m to 0.37 m, which demonstrates the substantial impact of antenna jitter on mitigating multipath errors across various satellite types.

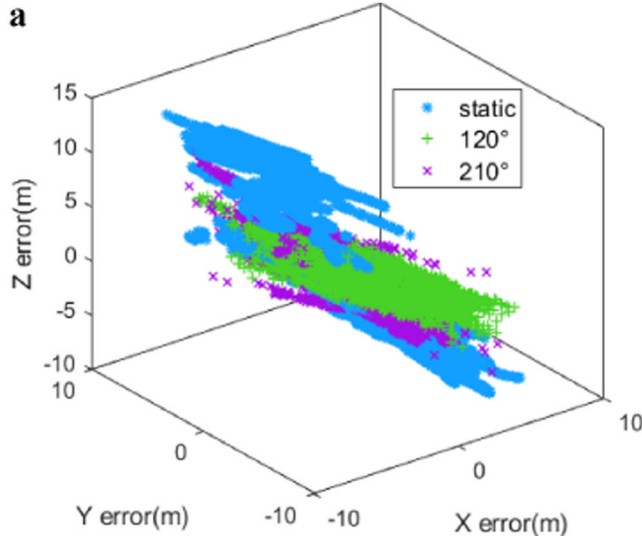

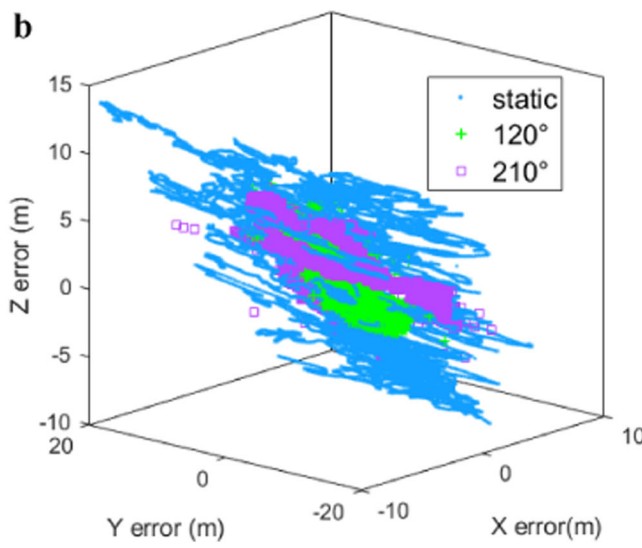

**Fig. 5 | 3D positioning error in different platforms. a** Open platform, and **b** Complex platform. The figure shows the comparison of the jitter and static antenna point positioning error, where (**a**) and (**b**) illustrate the open platform and complex platform, respectively. From the figure, it is evident that antenna jitter substantially enhances positioning accuracy by effectively mitigating multipath errors, leading to more convergent positioning errors (green and purple points), particularly in complex platforms. Specifically, in an open platform, the RMS error can be decreased from 2.45 m to 1.51 m. This reduction becomes even more pronounced in complex environments, where the RMS error is further diminished from 2.60 m to 0.98 m, achieving even higher positioning accuracy under such conditions.

### Single-point positioning error

Furthermore, we evaluate the actual positioning accuracy of the jitter antenna, where Fig. 5 shows the comparison of the jitter and static antenna point positioning error. Additionally, Tables 2 and 4 provide specific data on single-point positioning errors in open and complex platforms, respectively.

From Fig. 5, it is evident that antenna jitter substantially enhances positioning accuracy by effectively mitigating multipath errors, leading to more convergent positioning errors (green and purple points), particularly in complex platforms. Specifically, in an open platform setting, the RMS error can be decreased from 2.45 m to 1.51 m. This reduction becomes even more pronounced in complex environments, where the RMS error is further diminished from 2.60 m to 0.98 m, achieving even higher positioning accuracy under such conditions. Furthermore, the RMS error associated

**Table 4 | Result of single-point positioning Root Mean Square (RMS) and Maximum error in complex platform**

| Antenna type | RMS error (m) | Maximum error (m) |
|---|---|---|
| Static | 2.60 | 26.77 |
| 120° | 0.98 | 11.28 |
| 210° | 1.11 | 13.73 |

with different antenna jitter directions, within the complex platform, is negligible, being only 2 cm. This indicates that the direction of the antenna jitter exerts a minimal influence on positioning accuracy. Consequently, opting for linear jitter over rotational jitter not only conserves physical space but also maintains comparable levels of positioning accuracy.

To further ascertain the generalizability of the proposed method, this segment analyzes the 3D positioning error with vehicular antennas, which was compared with a static monitoring antenna. The product information of the vehicle antenna can be found in Supplementary Note 2.

Figure 6 presents the positioning error of different antennas, where the blue represents a stationary high-precision antenna, and the purple represents a jittering standard vehicle antenna. It becomes evident that the vehicular antenna manifests a decisive superiority with respect to cost and portability from Supplementary Note 2. Table 5 shows the result of single-point positioning error in complex platform with different antenna. From Table 5, we can see that the discrepancy in the 3D positioning accuracy of the jittering vehicle antenna, when compared to the monitoring antenna, is observed to be 1.38 meters. In the context of 2D positioning, the divergence is a mere 0.26 meters, indicating a negligible difference in the precision of location.

Furthermore, empirical data analysis reveals that the carrier-to-noise ratio (CNR) of the vehicular antenna is decremented by 7-10 dB relative to that of the monitoring antenna. This divergence is attributed to the antenna gain, which could lead to a reasonable inference. Therefore, if the CNR were equivalent, the ordinary antenna could fully supplant the costly and complex antenna when subjected to the proposed method.

### Conclusion

This paper proposes an effective method for mitigating multipath errors by using a jittering antenna. This model is supported by both theoretical analysis and real GNSS data, which validate the effectiveness of the proposed method. On one hand, theoretical analysis proves the mathematical feasibility of jitter. On the other hand, real data confirms its efficacy in suppressing multipath errors and its resilience to environmental variations. Furthermore, the results underscore the universality of the jitter direction, enhancing its security and practicality. However, the impact of antenna jitter speed on positioning accuracy and the effect of dynamic conditions such as in-vehicle antennas remain unproven, which will be our next focus. More importantly, we aim to substitute the choke ring antenna with a jitter antenna for low-cost, high-precision measurements.

### Methods

This chapter provides a detailed description of the antenna jitter method employed in this paper, including the model of antenna jitter and the minimum jitter amplitude for scenarios involving a single and multiple reflective surfaces.

### Antenna jitter model

In the field of satellite navigation, multipath signals can superimpose direct signals, leading to distortions in the correlation function. When the relative positions of the satellite and the receiver remain constant or change slowly over a long period, the multipath errors become difficult to mitigate through long-term observational averaging. Fortunately, the antenna jitter can rapidly alter the phase between the direct and multipath signals, converting multipath errors into pseudo-random errors. However, this also results in estimation biases for the direct signal. Therefore, this section will model the

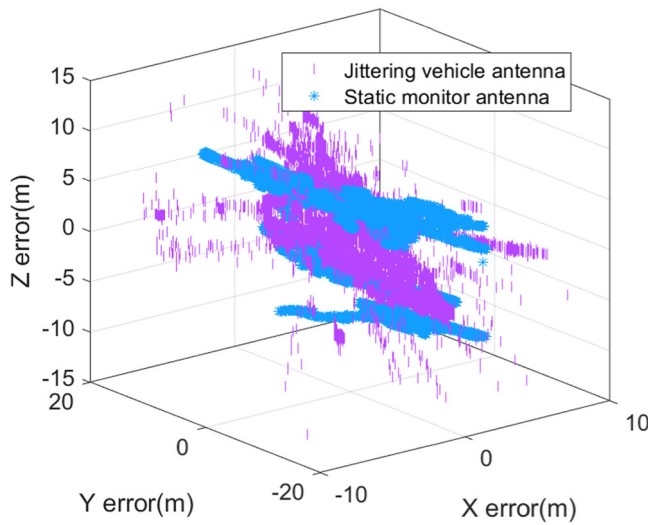

**Fig. 6 | 3D positioning error of different antennas.** The figure presents the positioning error of different antennas, where the blue represents a stationary high-precision antenna, and the purple represents a jittering standard vehicle antenna. It is evident that the vehicle antenna manifests a decisive superiority with respect to cost and portability from Supplementary Note 2. The discrepancy in the 3D positioning accuracy of the jittering vehicle antenna, when compared to the monitoring antenna, is observed to be 1.38 meters. In the context of 2D positioning, the divergence is a mere 0.26 meters, indicating a negligible difference of location precision. The results indicate that the jittering vehicle antenna achieves accuracy comparable to that of high-precision antennas, reducing the cost of the antenna from thousands of dollars to just a few dollars.

**Table 5 | Result of single-point positioning error in complex platform with different antenna**

| Antenna type | 3D error (m) | 2D error (m) |
|---|---|---|
| Vehicle | 4.01 | 2.03 |
| Monitor | 2.63 | 1.77 |

antenna jitter to determine how to minimize the impact on the direct signal while reducing multipath errors.

**The impact of antenna jitter on multipath signals.** If there is a wall or building around a GNSS antenna, the antenna receives multiple delayed signals, such as reflection and diffraction signals. In general, the presence of multipath signals is unavoidable, as a result, the signal received at the GNSS antenna is a combination of the direct and reflected signals[38].

$$
\begin{aligned}
x(t) &= As(t - \tau_0)\cos(2\pi f_c t + \theta_0) \\
&+ A\sum_{i=1}^{M} a_i s(t - \tau_0 - \Delta\tau_i)\cos(2\pi f_c t + \theta_0 + \Delta\theta_i)
\end{aligned}
\tag{2}
$$

where $s(t)$ denotes the GNSS spreading code sequence. $A, f_c, \tau_0$, and $\theta_0$ are amplitude, carrier frequency, time delay and carrier phase of the direct signal, respectively. $a_i, \Delta\tau_i$, and $\Delta\theta_i$ denote the power ratio, time delay, and carrier phase of the $i$-th multipath signal relative to the direct signal, respectively. In the GNSS code tracking loop, we consider a simple early-prompt-late (EPL) correlator and a simple type of code tracking loop employs early-minus-late discriminators. The discriminator function $D$ can be expressed as

$$
D = R(\hat{\tau}_0 - \tau_0 + d/2) - R(\hat{\tau}_0 - \tau_0 - d/2)
\tag{3}
$$

where $\hat{\tau}_0$ is the estimate of the direct signal time delay, $d$ is the early-late correlator spacing. $R(\cdot)$ is a triangle-shaped autocorrelation function

given by

$$
R(\tau) = \begin{cases} -\frac{1}{T_c}|\tau| + 1, & 0,, |\tau|,, T_c \\ 0, & \text{otherwise} \end{cases}
\tag{4}
$$

where $T_c$ is chip length, and it is assumed to be 1 in this study. Equation (4) is a standard case (such as PRN01 in GPS). It is important to note that in GPS, there are other PRN codes with different autocorrelation functions. However, we usually use the standard case as an example to provide an analysis framework[39], and the methods presented can be extended to other cases, provided the autocorrelation function is symmetric.

In the absence of multipath signals, the code delay error is 0, i.e. $\varepsilon = \hat{\tau}_0 - \tau_0 = 0$, when discriminator function $D = 0$. However, the signal autocorrelation function can be distorted by delayed reflection and diffraction signals. As an illustration, we consider the case that the direct signal is affected by the single reflected signal. Supplementary Fig. 2 illustrates the GNSS signal autocorrelation function in the two cases of different relative phases, where the multipath amplitude ratio $a_1$ is 0.5, the E-L spacing $d = 0.7$ chip, and the multipath delay $\Delta\tau_1$ is 0.5 chip.

As shown in Supplementary Fig. 2, it is apparent that the relative phase of the multipath signal is in-phase or anti-phase, and the maximum multipath error is $\varepsilon_{\max} = \pm a_1 d/2$. Furthermore, it should be noted that the error is positive when the relative phase ranges from 0 to 90 degrees, and negative when the relative phase ranges from 90 to 180 degrees, which allows us to randomize the multipath error by manipulating the relative phase. Although it is difficult to obtain an analytical expression cases including two or more multipath signals, when the reflected signal is smaller in amplitude than the direct signal, the expression for the extreme value of the error can be expressed as

$$
\varepsilon_{\max} = \frac{d}{2}\max\left[\sum_{i=1}^{M} a_i\cos(\theta_e + \Delta\theta_i)\right],
\tag{5}
$$

where $\theta_e = \theta_0 - \hat{\theta}_0$ represents the difference between the carrier phase of the direct signal $\theta_0$ and the local carrier phase estimate $\hat{\theta}_0$, i.e. carrier phase estimation error. As can be seen from the equation above, the maximum value of the error caused by multipath is related to the correlation interval $T_c$, the power ratio $a_i$, and the reflected wave phase $\Delta\theta_i$. When the antenna is static, a GNSS receiver is susceptible to multipath interference, which results in a large multipath error. In contrast, the relative carrier phase between the direct and the reflected signal changes rapidly when a GNSS antenna is jittering. As a result, the code range error depicted in will manifest as pseudo-random noise, and can be eliminated by averaging over a short period. We clarify the geometry relationship between antenna jitter and multipath error through Fig. 3.

As shown in Supplementary Fig. 3, $\alpha$ is the angle of the reflective surface relative to the horizontal surface, where various angles of reflection correspond to different scenarios (ground, buildings, slopes, etc.), and $\beta$ is the elevation angle of the satellite. The perpendicular distance from the antenna phase center $O$ to the reflective surface is $D_0$. Then, we establish a polar coordinate system making $O$ as the origin and the vertical line from $O$ to the reflecting surface as the polar axis. Hence, the position of the jitter antenna phase center $O'$ can be determined by the distance $r_n$ from $O'$ to $O$ and the angle $\psi_n$ between $O'O$ and the polar axis at the sample moment $n$, at which the relative time delay $\Delta\tau_1(r, \psi)$ and relative phase $\Delta\theta_1(r_n, \psi_n)$ are respectively computed as

$$
\begin{aligned}
\Delta\tau_1(r_n, \psi_n) &= 2(D_0 - r_n\cos\psi_n)\sin(\beta + \alpha)/c \\
&= \Delta\tau_1 - 2r_n\cos\psi_n\sin(\beta + \alpha)/c,
\end{aligned}
\tag{6}
$$

$$\Delta\theta_1(r_n, \psi_n) = 4\pi(D_0 - r_n \cos\psi_n)f_c \sin(\alpha+\beta)/c$$
$$= \Delta\theta_1 - 4\pi r_n \cos\psi_n f_c \sin(\alpha+\beta)/c, \qquad (7)$$

where $\Delta\tau_1 = 2D_0 \sin(\beta+\alpha)/c$ and $\Delta\theta_1 = 4\pi D_0 f_c \times \sin(\alpha+\beta)/c$ denote the relative time delay and relative phase when the antenna is stationary, respectively. $c$ represents the speed of light in a vacuum. Note that this method may not make changes in path difference between a direct and a reflected signal when $\psi_n = 90^\circ$or$270^\circ$(parallel to the reflective surface).

Ultimately, based on the correlator characteristics of the GNSS receiver, the multipath error at each moment $n$ when the antenna is jittering can be expressed as Eq.(8).

$$\varepsilon(a_1, \Delta\tau_1, \Delta\theta_1, r_n, \psi_n) =$$
$$\begin{cases} \frac{\Delta\tau_1(r_n,\psi_n)a_1 \cos(\theta_e+\Delta\theta_1(r_n,\psi_n))}{1+a_1 \cos(\theta_e+\Delta\theta_1(r_n,\psi_n))}, 0 < \Delta\tau_1(r_n,\psi_n) \le \tau_p, \\[2mm] \frac{a_1 d \cos(\theta_e+\Delta\theta_1(r_n,\psi_n))}{2a_0}, \tau_p < \Delta\tau_1(r_n,\psi_n) \le \tau_q, \\[2mm] \frac{a_1 \cos(\theta_e+\Delta\theta_1(r_n,\psi_n))}{2-a_1 \cos(\theta_e+\Delta\theta_1(r_n,\psi_n))} \\[2mm] \times \left(T_c + \frac{d}{2} - \Delta\tau_1(r_n,\psi_n)\right), \tau_q < \Delta\tau_1(r_n,\psi_n) \le T_c + \frac{d}{2}, \\[2mm] 0, \Delta\tau_1(r_n,\psi_n) > T_c + \frac{d}{2}, \end{cases} \qquad (8)$$

where $\tau_p = [1 + a_1 \cos(\theta_e(r,\psi) + \Delta\theta_1(r,\psi))]d/2$, $\tau_q = T_c + a_1 d \cos(\theta_e(r,\psi) + \Delta\theta_1(r,\psi))/2$. $\tau_p$ and $\tau_q$ represent the threshold points at which the $\Delta\tau_1(r_n, \psi_n)$ affects the $\varepsilon(a_1, \Delta\tau_1, \Delta\theta_1, r_n, \psi_n)$, primarily influenced by $d$. $\varepsilon(a_1, \Delta\tau_1, \Delta\theta_1, r_n, \psi_n)$ will exhibit a uniform distribution with a mean of zero when $\tau_p < \Delta\tau_1(r_n, \psi_n) < \tau_q$ for different $\Delta\theta_1$, which can be completely compensated with the method proposed in this paper. As depicted in Eq.(8), the multipath error has been transformed into a rapid-change error when the antenna is jittering. It can be also observed that any jitter motion of an antenna (linear, rotational, random, etc.) can be projected into variables related to the amplitude $r_n$ and angle $\psi_n$ of linear motion. Therefore, the following text mainly takes linear motion as an example for analysis.

Note that, the propagation path of the direct signal to the phase center of the antenna also changes due to antenna jitter. According to the geometric relationship, the time delay of the direct signal at the sample moment $n$ is

$$\tau_0(r_n, \psi_n) = \tau_0 + r_n \sin(\beta+\psi_n+\alpha)/c \qquad (9)$$

As a result, the time delay at each moment $n$ of the direct signal is estimated as

$$\hat{\tau}_0(r_n, \psi_n) = \tau_0(r_n, \psi_n) + \varepsilon(a_1, \Delta\tau_1, \Delta\theta_1, r_n, \psi_n) \qquad (10)$$

Following the error correction based on Eq.(9), the code delay estimate for various sample moments $n$ is

$$\hat{\tau}_0(n) = \tau_0(r_n, \psi_n) - r_n \sin(\beta+\psi_n+\alpha)/c$$
$$+ \varepsilon(a_1, \Delta\tau_1, \Delta\theta_1, r_n, \psi_n) \qquad (11)$$
$$= \tau_0 + \varepsilon(a_1, \Delta\tau_1, \Delta\theta_1, r_n, \psi_n)$$

where $\tau_0$ denotes the real time delay. Overall, it is the fast time-varying state of $(r_n, \psi_n)$ that fundamentally determines whether the multipath error can successfully transform from a "Standing Multipath" to a "Pseudo-Random" error.

**The impact of antenna jitter on direct signals.** The "Pseudo-Random" multipath errors can be effectively reduced through averaging. However, Eq. (11) highlights that while we may achieve precise control over the antenna jitter angle and amplitude, the accurate determination of the satellite elevation angle and the tilt of the reflective surface is not practical. Consequently, it is not viable to correct the estimated bias of the direct signal with precision at every instant. Nevertheless, if the biases conform to a specific statistical distribution, they can be eliminated by employing the same averaging technique which was applied to multipath errors.

The estimated bias of the direct signal is denoted as $\delta_n = r_n \sin(\beta+\psi_n+\alpha)$, and after the averaging process, the bias can be represented as

$$\delta = \frac{1}{N}\sum_{n=1}^{N} r_n \sin(\beta+\psi_n+\alpha) \qquad (12)$$

If the reflective surface angle and the satellite elevation angle remain constant, the distribution of the direct signal bias is entirely contingent upon the jitter mechanism. In light of this, provided that the jitter is uniform, the estimated bias of the direct signal will conform to a uniform distribution, with the bounds of this distribution being determined by the maximum deviation amplitude, i.e. $r_n \sin(\beta+\psi_n+\alpha) \sim U(\delta_-, \delta_+)$, where:

$$\delta_+ = \max(r_n \sin(\beta+\psi_n+\alpha))$$
$$\delta_- = \min(r_n \sin(\beta+\psi_n+\alpha)) \qquad (13)$$

Then, the mean and variance of the error after averaging can be represented as:

$$E(\delta) = (\delta_+ + \delta_-)/2 \qquad (14)$$

$$D(\delta) = \frac{1}{N^2}D\left[\sum_{n=1}^{N} r_n \sin(\beta+\psi_n+\alpha)\right]$$
$$= \frac{(\delta_+ - \delta_-)^2}{12N} \qquad (15)$$

In Eq. (15), it is evident that as long as the jitter is symmetrical (with the trajectory being point-symmetric with respect to the stationary point), the mean of the error post-averaging is zero, thus bias can be cancelled. Additionally, an increase in the number of samples results in a reduction of the variance. For ease of presentation, a simulation experiment of linear perpendicular jitter relative to the reflective surface is undertaken. The simulation parameters are delineated in Supplementary Table 1, with the maximum jitter amplitude of the linear specified as:$r_M = \max(r_n)$. The specific results are shown in Supplementary Fig. 4.

It can be observed in Supplementary Fig. 4 that bias is fully negated upon averaging with symmetric jitter. Conversely, the bias post-averaging remains unmitigated with asymmetric jitter. Thus, adjustments based on the jitter trajectory are required when symmetric jitter cannot be implemented, yet achieving precise corrections is notably difficult. On the other hand, the variance of the direct signal bias decreases progressively as the number of samples increases. Once the number of samples exceeds 1000, the variance reduction exhibits a linear trend and below $10^{-5}$ m, meeting the requirements for real-time high-precision positioning.

**Minimum antenna jitter amplitude**
This section delves further into the mitigation of multipath errors. Drawing from statistical principles, it is understood that the efficacy of estimation post-averaging is contingent upon the comprehensiveness of the underlying space. To be more precise, an increase in the jitter amplitude yields an estimation that more closely approximates the true value upon averaging. However, it is imperative to consider the practical implications, as an excessive amplitude of jitter can engender many detrimental effects. Therefore, in the quest to find an effective minimum jitter amplitude, this chapter initially examines the impact of jitter amplitude and angle with a single reflective surface. Building upon this foundation, we proceed to analyze the interplay among multiple multipath components in the presence of several reflective surfaces.

**Single reflective surface.** Building on our earlier analysis, finding the minimum jitter amplitude for a single reflective surface involves solving

the following optimization problem.

$$\arg \min \frac{1}{N} \sum_{n=1}^{N} \varepsilon(a_1, \Delta\tau_1, \Delta\theta_1, r_n, \psi_n),$$
$$\arg \min r_M. \tag{16}$$

where $N$ represents the quantity of co-sampling points within an observation period. Through a quantitative analysis of the relationship between multipath error averaged and $(r_n, \psi_n)$, we ascertain the minimum amplitude of jitter when solely one reflective surface exists, which is outlined in Theorem 1.

**Theorem 1**: The minimum jitter amplitude is affected by the jitter angle. When only one multipath reflective surface exists, the jitter amplitude that optimally utilizes minimal space for achieving the most effective suppression can be expressed as

$$r_{M(opt)} = \frac{c}{4f_c |\sin(\alpha + \beta) \cos\psi_n|}. \tag{17}$$

**Remark 1**: Theorem 1 assumes that the minimal jitter amplitude is influenced by the relative positioning of the reflective surface in relation to the antenna. Specifically, the optimal jitter direction is along the normal of the reflecting surface in the presence of a single fixed reflecting surface. In contrast, a rotating antenna appears to offer a superior effect on multipath signals when multiple distinct reflective surfaces are present in theory, as it can account for reflections from all directions[37]. However, subsequent experiments demonstrate that linear jitter can effectively eliminate errors with almost the same performance compared to a rotating antenna, as the randomly distributed reflective surfaces in the real world can counteract the effect of jitter direction.

Moreover, it is important to emphasize that the actual minimum jitter amplitude needs to be determined by comparing it with the random error component in the pseudorange measurements. However, this paper aims to transform multipath errors into random errors and explore the impact of antenna jitter patterns on error transformation. Therefore, random errors of the pseudorange have been temporarily neglected to demonstrate the impact of antenna jitter. A detailed discussion of random errors will be discussed in our future work.

**Proof**: Firstly, we observe that Eq. (8) is a piecewise function, yet different segments possess symmetry. Consequently, it is beneficial to analyze the error function of the first segment as an illustrative example. Although it is evident that the error function is influenced by $\Delta\tau_1(r_n, \psi_n)$ and $\Delta\theta_1(r_n, \psi_n)$, the small change of $\Delta\tau_1(r_n, \psi_n)$ because of the antenna's minor-jitter. Therefore, the multipath error can be approximated as a periodic function, thus the first subsection of Eq. (8) can be reduced to

$$\varepsilon(\Delta\theta_1, r_n, \psi_n) = \frac{a_1 \Delta\tau_1 \cos(\theta_e + \Delta\theta_1(r_n, \psi_n))}{1 + a_1 \cos(\theta_e + \Delta\theta_1(r_n, \psi_n))} \tag{18}$$

Based on calculus and statistics, the averaging multipath error is equivalent to taking the expected value of the multipath error over the period when the sampling number is enough. Consequently, the multipath error can be simplified as:

$$\varepsilon = \frac{1}{N} \sum_{n=1}^{N} \varepsilon(\Delta\theta_1) = \frac{c\Delta\tau_1}{8\pi r f_c \sin(\alpha + \beta) \cos\psi_n}$$
$$\times \int_{\Delta\theta_l}^{\Delta\theta_s} \varepsilon(\Delta\theta_1, r_n) d\Delta\theta_1(r_n) \tag{19}$$
$$= \frac{c\Delta\tau_1}{\Delta\theta_s - \Delta\theta_l} [F(\Delta\theta_s) - F(\Delta\theta_l)]$$

where the upper and lower bounds of the integral can be expressed as $\Delta\theta_s = \Delta\theta_1 + 4\pi r_M f_c \sin(\alpha + \beta)/c$ and $\Delta\theta_l = \Delta\theta_1 - 4\pi r_M f_c \sin(\alpha + \beta)/c$, respectively, which represent the relative phase at the maximum

deviation i.e. $r_M = \max_{1 \le n \le N} r_n$. Eq. $F(x) = x - \log([a_1 + \cos(x) + \sqrt{a_1^2 - 1} \times \sin(x)]/[a_1 \cos(x) + 1])/\sqrt{a_1^2 - 1} = x - G(x)$ represents the integral original function of the first stage, where $G(x)$ denotes the periodic part of $F(x)$. As a result, Eq. (19) can be calculated by transforming the upper and lower bounds of the integral into a periodicity of $G(x)$. However, the transformation process across the breakpoints will increase the integral value by one cycle, which can be calculated as

$$\kappa_1 = F(3\pi) - F(\pi). \tag{20}$$

Therefore, the multipath error Eq. (19) can be further simplified as

$$\varepsilon = \frac{c\Delta\tau_1}{8\pi r_M f_c \sin(\alpha + \beta)}$$
$$\times [W_2 - W_1 - (G(W_1) - G(W_2)) + p\kappa], \tag{21}$$

where $W_1 = \mathrm{mod}(\Delta\theta_l - \pi, 2\pi) + \pi$, $W_2 = \mathrm{mod}(\Delta\theta_s - \pi, 2\pi) + \pi$, and $p$ denotes the number interruption point is crossed, which can be calculated as $p = (\Delta\theta_l - W_1 + W_2 - \Delta\theta_s)/(2\pi)$.

Although it is evident that $p$ is linked to both $\Delta\theta_l - \Delta\theta_s$ and $W_1 - W_2$, $\Delta\theta_l - \Delta\theta_s$ can be canceled without of parentheses, so that Eq. (21) will reach its minimum value when $W_1 = W_2$, $G(W_1) = G(W_2)$, which can be expressed mathematically as (Although the integrals of one period of Eq. (10) are not zero, the selection of the integrals of the whole period ensures that the antenna jitter obtains all the relative phase values, which is highly approximate to 0.)

$$\mathrm{mod}(\Delta\theta_s - \Delta\theta_l, 2\pi) = 0 \tag{22}$$

From Eq. (22) we can see that the jitter amplitude $r_M$ to achieve the best suppression effect of multipath error must satisfy

$$r_M = k \frac{c}{4f_c |\sin(\alpha + \beta) \cos\psi_n|}, k = 1, 2, \cdots \tag{23}$$

What stands out in Eq.(23) is that the bigger jitter amplitude is not necessarily more effective, due to there being a certain periodicity in the effect of suppressing the multipath error. Therefore, in this study, the minimum jitter amplitude can be achieved by choosing $k = 1$ to satisfy the optimal suppression effect.

The other piecewise follows a similar pattern as described above. To validate the accuracy of the jitter amplitude and angle selection, the Supplementary Fig. 5 illustrates the variation trend of multipath error versus jitter amplitude and angle. Here, the multipath delay $\Delta\tau_1$ is 0.01 chip, the multipath power ratio $a_1$ is 0.5, the satellite elevation angle $\beta$ is 30°, the reflective surface inclination $\alpha$ is 90°, and the carrier frequency is 1561.98 MHz.

It can be seen from the comparison in Supplementary Fig. 5 that the multipath error decreases different values with the various jitter amplitude and angle, where the jitter angle and amplitude of minimum error are strongly consistent with the theoretical analysis. The above jitter angle and amplitude are equivalent to the antenna jittering ±5 cm along the direction normal to the reflecting surface, such a jitter amplitude is acceptable in the real environment. Additionally, it should be noted that multipath error diverges when jitter angles are $\pi/2$ and $3\pi/2$, i.e., the jitter directions are parallel to the reflecting surface. This also confirms the derivation in Theorem 1, which emphasizes the need to select a reasonable angle.

amplitude and angle.

To summarize, the analytical solution of the multipath error when the antenna is jittered in the way of theorem 1 can be given by the following theorem.

**Theorem 2**: When only one reflecting surface exists, the antenna jitter in the method of Theorem 1 can reduce the multipath error to

$$\varepsilon(a_1, \Delta\tau_1) = \begin{cases} \frac{\Delta\tau_1 \kappa_1}{2\pi}, 0 < \Delta\tau_1 \leq \tau_{p1}, \\ \frac{(a_1-1)\kappa_1 \Delta\tau_1}{4\pi a_1} + \frac{(1-a_1^2)d\kappa_1}{8\pi a_1}, 0 < \Delta\tau_1 \leq \tau_{p2}, \\ 0, \tau_{p2} < \Delta\tau_1 \leq \tau_{q1}, \\ \frac{(a_1-2)\kappa_2}{4\pi a_1}[\Delta\tau_1 - T_c + \frac{(a_1+1)d}{2}], \tau_{q1} < \Delta\tau_1 \leq \tau_{q2}, \\ (T_c + \frac{d}{2} - \Delta\tau_1)\frac{\kappa_2}{2\pi}, \tau_{q2} < \Delta\tau_1 \leq T_c + \frac{d}{2}, \\ 0, \Delta\tau_1 > T_c + \frac{d}{2}, \end{cases} \quad (24)$$

where $\tau_{p1} = (1-a_1)d/2$, $\tau_{p2} = (1+a_1)d/2$ $\tau_{q1} = T_c - (a_1+1)d/2$, and $\tau_{q2} = T_c + (a_1-1)d/2$. $\kappa_1$ and $\kappa_2$ represent the integral value for one period of the first and third piecewise of Eq. (8), respectively.

**Remark 2**: Theorem 2 illustrates that even if the relative position of the satellite and the GNSS receiver remains constant or changes slowly, the multipath error will be converted into "pseudo-random" noise when the antenna is jittering, at which the average error will transform into a simple linear function. This transformation is beneficial for mitigating the short-delay multipath, which traditional methods are challenging to eliminate. Moreover, the multipath error at most of the delay will be canceled to zero if the narrow correlation technique is combined simultaneously. Unless otherwise specified, the simulation conditions used here are the same as those depicted in Supplementary Fig. 5. Supplementary Fig. 6 illustrates the variation trend of multipath error versus jitter amplitude for different time delays, where "Sim" refers to the simulated results obtained through numerical simulations, while "Ana" refers to the analytical results, which are derived from the theoretical analysis in the paper.

As can be seen from Supplementary Fig. 6, it is evident that the multipath error does not monotonically decrease, as the jitter amplitude increases. Instead, it reaches different minimum values at $k = 1, 2, 3,$ and4, which validates the accuracy of Theorem 2 under conditions of short time delay. However, multiple multipath signals with diverse time delays are encountered in practical scenarios. Thus, assessing the efficacy of antenna jitter in reducing multipath errors necessitates consideration of all time delays, typically represented using multipath error envelopes. Supplementary Fig. 7 illustrates the multipath error envelopes of the 1-chip correlation distance.

From Supplementary Fig. 7 we can see that the jitter antenna exhibits an obvious suppression effect on multipath errors across all time delays, where the peak value and effective area of the multipath error envelope are notably reduced. Specifically, the peak error can be reduced by 66% when employing half of the optimal jitter amplitude, and the peak error has been reduced by 66 m compared to the static antenna when employing the optimal jitter amplitude. Furthermore, the theoretical and simulation results are in excellent agreement, which comprehensively verifies the correctness of Theorem 2. To verify the combined performance of the methods presented in this paper with other approaches, Supplementary Fig. 8 validates the case when the antenna jitter is combined with narrow correlation technology.

In Supplementary Fig. 8, there is a clear downward trend of multipath when combined with narrow correlation technology. In particular, the envelope peak value of the correlation distance of 1/8 chip can be further reduced by 4/5, and the effective length of the envelope can be reduced by 7/8 compared to the correlation distance of 1/2 chip. As a result, the envelope peak value can be reduced by 98%, equivalent to around 0.005 chips. Moreover, approximately 87.5% of the overall time delay of the multipath error can be minimized to approach zero when random errors in the pseudorange measurements are not considered, indicating the theoretical effectiveness of the antenna jitter.

**Multiple reflective surfaces**.

$$\varepsilon(a_i, \Delta\tau_i, \Delta\theta_i, r_n, \psi_n)$$
$$= \begin{cases} \sum_{i=1}^{M} \frac{\Delta\tau_i(r_n, \psi_n)a_i \cos(\theta_e + \Delta\theta_i(r, \psi))}{1 + a_i \cos(\theta_e + \Delta\theta_i(r, \psi))}, 0 < \Delta\tau_i(r, \psi) \leq \tau_p \\ \sum_{i=1}^{M} \frac{a_i d \cos(\theta_e + \Delta\theta_i(r_n, \psi_n))}{2a_0}, \tau_p < \Delta\tau_i(r, \psi) \leq \tau_q \\ \sum_{i=1}^{M} \frac{a_i \cos(\theta_e + \Delta\theta_1(r_n, \psi_n))}{2 - a_i \cos(\theta_e + \Delta\theta_1(r_n, \psi_n))} \\ \times (T_c + \frac{d}{2} - \Delta\tau_i(r_n, \psi_n)), \tau_q < \Delta\tau_i(r_n, \psi_n) \leq T_c + \frac{d}{2} \\ 0, \Delta\tau_i(r_n, \psi_n) > T_c + \frac{d}{2} \end{cases} \quad (25)$$

In the preceding section, a quantitative analysis and simulation verification were conducted to ascertain the optimal jitter paths under the influence of a signal reflective surface. However, the sources of multipath in real-world environments are more extensive. Consequently, this section extends the investigation to account for the presence of multiple reflective surfaces. The analysis and resolution of this more complex situation aim to provide a robust theoretical framework, that offers explanatory depth and practical guidance for experiments.

When $M$ reflective surfaces are present, they give rise to $M$ distinct multipath signals. Consequently, the multipath error can be articulated by Eq.(25).

At this point, the averaged multipath error can be expressed as:

$$\varepsilon = \sum_{i=1}^{M}[\frac{1}{N}\sum_{n=1}^{N}\varepsilon(a_i, \Delta\tau_i, \Delta\theta_i, r_n, \psi_n)] \quad (26)$$

In practical applications, obtaining precise angular calibration of multiple reflective surfaces is typically unfeasible. Nonetheless, the optimal jittering strategy tailored for a single reflective surface can remain highly effective, particularly when a dominant reflective surface is present—one that exhibits a substantially greater reflection amplitude than other transmitting surfaces, such as the closest building. To this end, we have extended our analysis with additional simulations aimed at delineating the interplay among the number of reflective surfaces, antenna jitter, and multipath errors.

We initiate our study with a satellite elevation angle of 30°. The reflective surface with the predominant reflection amplitude is designated as the main reflective surface and the position was determined. For the remaining reflective surfaces, the reflection amplitude, time delay, and inclination are assigned random values within the parameters of typical values. The specific simulation parameters are delineated in Supplementary Table 2.

We introduce the parameters of the main reflective surface into the framework established in Section 3.1 to deduce the optimal jitter direction and amplitude. Subsequently, we conduct 1000 Monte Carlo simulations under three conditions: static antenna, optimal jitter, and random directional jitter. Supplementary Fig. 9 presents the mean and mean square (MS) deviation across varying counts of reflective surfaces.

The comparison of Supplementary Fig. 9 reveals that the optimal jitter mode, previously determined for a single reflective surface, remains effective even when multiple reflective surfaces are present. Specifically, the MS error associated with the jitter antenna is markedly lower than the static antenna, and this discrepancy tends to increase with the addition of reflective surfaces. Furthermore, the optimal jitter method yields markedly smaller multipath errors than other antenna states, because it accounts for the calibration of the primary reflective surface. This suggests that the optimal jitter direction and angle play an important role in real scenarios if the primary reflective surface can be calibrated.

However, it is also possible to encounter a situation devoid of a main reflective surface, with no obvious distinction in the reflection amplitude among the reflective surfaces. In such cases, we select a reflective surface at random for to determine the jitter amplitude and angle. Subsequently, we compare the performance of this calibrated setup with that of a rotating

**Table 6 | Result of multipath Mean Square (MS) and range error in the open platform**

| Antenna type | PRN = 1 | | PRN = 59 | |
|---|---|---|---|---|
| | MS error (m) | Range error (m) | MS error (m) | Range error (m) |
| Static | 0.33 | 2.08 | 0.51 | 2.84 |
| 120° | 0.15 | 0.76 | 0.13 | 0.72 |
| 210° | 0.16 | 1.95 | 0.17 | 1.16 |

antenna at the same amplitude, as well as with a system that employs a random jitter amplitude but the same direction. The specific simulation parameters for these comparisons are detailed in Supplementary Table 3 and the results can be found in Supplementary Fig. 10.

It can be observed that an increase in jitter amplitude primarily results in a reduction of variance within scenarios featuring a higher number of reflective surfaces in Supplementary Fig. 10. Conversely, the mean value of the multipath error tends to increase, which underscores the efficacy of an optimally determined jitter amplitude. Furthermore, the discrepancy between the multipath errors generated by the optimal jitter method and those produced by a rotating antenna is found to be minimal. This finding validates that linear jitter is equally effective as the rotating antenna technique multiple reflective surfaces, with the added benefit of conserving space.

To further compare the performance of the antenna jitter with other mainstream algorithms, we conducted simulations to assess the robustness and precision of each algorithm. The number of reflective surfaces was set to M = 6, with the angular positions of the reflective surfaces and the antenna jitter angles being selected by Table 6. The amplitude of the antenna jitter was determined based on Eq.(17) with an initial bias set at 0.2 chips, and the correlator spacing $d = 1/8$ chip. Considering the influence of random errors, we use signal-to-noise ratio ($SNR$) as the variable to observe the error suppression effect. The relationship between SNR and carrier-to-noise density ratio ($C/N_0$) is[40]:

$$SNR = C/N_0 - 10\lg(B_e T_{coh}) \tag{27}$$

where the unit of $C/N_0$ is dBHz, the unit of $SNR$ is dB, the receiver bandwidth $B_e$ was set to $B_e = 1$MHz, and the unit of coherent time $T_{coh}$ was set to $T_{coh} = 1$ms.

As shown in Supplementary Fig. 11, it is evident that the antenna jitter possesses distinct advantages in terms of robustness and efficacy. This technique demonstrates notable performance under low SNR conditions, achieving steady-state convergence 10–15 dB earlier than other methods, which is particularly suited to counteract the complexities of urban and other challenging environments. Furthermore, the error during the steady state of this method is reduced by approximately 80–90% compared to traditional narrow correlation and Early-Prompt-Late (EPL) methods, even surpassing the performance of more complex algorithms such as Code Correlation Reference Waveform (CCRW)[13] and Double-delta, which is attributed to the introduction of observables through motion. Thus, this method enables the direct application to conventional receivers as an independent parameter. This results in low-complexity, high-precision positioning and it can be integrated with other high-precision algorithms to meet the ultra-high-precision requirements of monitoring stations.

## Data availability
All data in this article can be obtained at: https://pan.baidu.com/s/1i-bN8vayXBIS4EuMT-w2UA?pwd=1234, the password is 1234. or https://drive.google.com/drive/folders/1kOVAo1Sep_RgjK7tJXIIypTad-Vt-teD.

## Code availability
The code presented in this study can be obtained at: https://gitcode.com/liujiayng/A_Multipath_Error_Cancellation_Method_Based_on_Antenna_Jitter/overview.

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

## Acknowledgements

This research was funded by the National Natural Science Foundation of China under Grant U20A20193 and 62201585. Furthermore, the authors would like to thank Dr. Zhibin Xiao, Dr. Pengpeng Li, and Mr. Kunmu Li from the National University of Defense Technology for the experimental collaboration.

## Author contributions

Jiyang Liu: He was responsible for most of the writing, experimentation, and revisions of this paper. Feixue Wang: He proposed the key innovations of the paper and determined the overall structure of the writing. Xiaomei Tang: She was responsible for addressing the technical challenges in the implementation of the paper, such as the specific derivation of the formulas. Sixin Wang: He was responsible for gathering related research and played a key role in the revision process. Muzi Yuan: He was responsible for the specific implementation of the experiments in this paper, including data collection, analysis, and comparison. All authors have read and agreed to the published version of the manuscript.

## Competing interests

The authors declare no competing interests.

## Ethics

This study adheres to ethical guidelines, ensuring inclusivity and transparency.
