## [Transparent Peer Review file · Communications Engineering]

A Multipath Error Cancellation Method Based on Antenna Jitter

Corresponding Author: Professor Feixue Wang

Version 0:

Reviewer comments:

Reviewer #1

(Remarks to the Author)

This manuscript discusses characteristics and effects of a linear antenna jitter method to cancel multipath errors, including simulation and observational results. Although it contains interesting approaches and results, there are some unclear points. Please revise the manuscript according to the following comments.

Conditional comments:

1. P2, Left A paragraph of "In addition, ...Wen et al., 2020)"

I couldn't correctly understand the meaning of this paragraph, including the structure of the sentence. Please clarify the structure and add an explanation.

2. P3, Left, Line 5, "choke antennas"

I think it should be "choke ring antennas". If so, please correct it.

3. P3, Right, Equation (3)

This autocorrelation function is a standard case (for example, PRN01 etc. in GPS). In the case of GPS, there are some other cases, so please mention this issue and add an explanation on how it will be handled in this study. In addition, "o.w." is not clear. Please explain it in other words.

4. P3, Right, Fig. 2

Please add a description of the Early/Late correlator spacing in this diagram. Also, since this diagram shows a special case where the position of either the Early or Late correlator coincides with the point where the rate of change changes, please add an explanation of what this special positional relationship means. Additionally, I think code phase error will be code range error. If so, please modify this.

5. P3, Right, Line 17, An equation of e_{\max}

Please add an explanation for variable d .

6. P3, Right, Line 28, "the carrier phase estimation error"

It is unclear what the carrier phase estimation error means, so please add some explanations.

7. P3, Right, Line 31, " T_c "

It is unclear because T_c is not described in Equation (4). Please modify it correctly.

8. P4, Left, Equation (5) and (6)

Please mention in the text that this method may not make changes in path difference between a direct and a reflected signals when Ψ is 90 degrees (parallel to the reflecting surface). Additionally, please add some explanations on c and f_c in the text.

9. P4, Right, Line 4, "Eq. (10)"

Although the equation numbers (10) are assigned, it should be better to follow the order in which they appear in the text. Also, please add some explanations of what τ_p and τ_q mean in this method.

10. P4, Right, Line 28, "Fig. 4"

I could not find the Fig 4, so please check and modify it.

11. P4, Right, Line 42, "10-510-5 meters"
Please modify it correctly.

12. P5, Right, Equation (13)
Please add some explanations on N and transformation of the variance formula.

13. P6, Left, Line 45, A sentence of "However, ... error."
Please add information on evidence from the previous research articles etc.

14. P7, Left, Fig. 7
Please mention in the text that when jitter angles are $1/2 * \pi$, $3/2 * \pi$, the jitter directions are parallel to the reflecting surface and it causes diverges. Also, I believe that the actual minimum jitter amplitude needs to be determined by comparing it with the random error component in the pseudorange measurements, so please add some explanations on this issue in the text.

15. P8, Left, Fig. 9
The figure contains the words "Sim" and "Ana". Please add some explanations on them, including the differences between them. The same would be for Fig10.

16. P8, Right, Line 5, A paragraph of "In Fig. 10, ... the antenna jitter."
Actually, it is needed to take into account random errors in the pseudorange measurements. Please mention it in this paragraph.

17. P10, Left, Fig. 13
Normally, a signal strength of more than about 26 dB in C/N0 is required to capture a GNSS signals. Please add some explanation on why signal tracking is possible even at 0 dB, including the correlator spacing set in this figure.

18. P10, Left, Line 13, "as shown in Figure 12"
I think it should be "as shown in Figure 13"

19. P10, Left, Line 23, "CCRW"
Please add description of the original words.

20. P10, Right, Line 32, "In Fig. 14, No. 1 and No. 59 satellites"
Please add core satellite name of them, i.e. Beidou.

21. P21, Left, Table 4
Please add product information on GNSS antenna and receiver in the Table or in the text.

21. P14, Left, Table 10
Please add product information on GNSS antennas in the Table or in the text.

22. P14, Left, Line 12, A sentence of "This divergence is attributed to the antenna gain, which leads to a reasonable inference."
One possibility could be remained that the antenna phase characteristic is not uniform in the vehicle antenna. Therefore, please modify this sentence as "This divergence is attributed to the antenna gain, which could lead to a reasonable inference." etc.

Other comments:

A1. P12, Left, Line 5, "Specifically, the B1"
-> Specifically, MS error for the B1

A2. P13, Left, Line 21, " where we choose the root ... 60 satellites."
This part would be not necessary.

A3. P13, Right, Line 9, "Table 6"
It would be removed.

Reviewer #2

(Remarks to the Author)

This manuscript proposes a method to offset GNSS signal multipath error by using antenna jitter, which is realized by rapidly changing signal phase, overcoming the limitation of traditional methods. The method can be integrated with other technologies and is suitable for a wide range of receivers. The analysis shows that the minimum jitter amplitude can effectively reduce the error in linear jitter. Experiments show that when the antenna jitter is $\pm 6.2\text{cm}$, the pseudo-distance mean square error is reduced by 72.42%, and the three-dimensional positioning error is reduced by 62.31%, which is significantly superior to the traditional method in robustness, effectiveness and applicability. The method proposed in this manuscript has certain reference value for reducing the influence of multi-path effect of satellite signal and improving the precision of satellite positioning

1. As mentioned in the abstract, The results show that an antenna jitter of $\pm 6.2\text{cm}$ can reduce the pseudo-range mean square error by 72.42% and the 3D localization error by 62.31% compared with a stationary antenna. This experimental result is only for a specific experimental situation, and does not have universal reference significance, so it is not suitable to appear in the abstract.
2. The manuscript describes the method of determining minimum jitter amplitude in a lot of length, but the process and results of determining minimum jitter amplitude are not reflected in the experimental part, and the basis for selecting an antenna jitter of $\pm 6.2\text{cm}$ is not clarified. This shows that there is a separation between the experimental process and the theoretical analysis.
3. There are typographical problems and writing errors in the article, for example,
 - a) We clarify the geometric relationship between the antenna jitter and multipath error through Fig. 3 The sentence lacks a period at the end.
 - b) There is a large amount of blank space in the page before formula (10) and formula (24).
 - c) There is a meaningless black block at the top of Figure 7.
 - d) Multiple tables in the article are disconnected, affecting reading.
 - e) The antenna parameters are delineated in the Table 9Table 6. E) The antenna parameters are delineated in the Table 9Table 6. The absence of and in the sentence above.
4. There are problems in the drawing quality of the pictures in the article Fig.7, Fig.19, Fig.20 images are blurred, and it is suggested to redraw Fig.14 and Fig.17.

Version 1:

Reviewer comments:

Reviewer #1

(Remarks to the Author)

I have confirmed that the authors have responded carefully to the reviewers' comments and that the manuscript has reflects them, however I believe the following point requires further revision:

1. Comments 16 of Reviewer 1:

The authors revised a paragraph of "In Fig. 9, ..." from line of 10 in the right part of page 8 in the manuscript. I think "the correlation distance of 1 chip" in line of 15 in the page should be "the correlation distance of 1/2 chips". If so, please modify it.

Reviewer #2

(Remarks to the Author)

In this manuscript, a novel multipath error cancellation approach utilizing antenna jitter is proposed. Simulations and experimental tests are performed to verify the effectiveness of the proposed method.

Responses to the Reviewers

Manuscript ID: COMMS-24-0484

Title: A Multipath Error Cancellation Method Based on Antenna Jitter

The authors would like to thank the reviewers for volunteering their time to review our paper and provide us with valuable comments. We have revised our manuscript based on the comments and suggestions. The following are our point-by-point responses to the raised comments, highlighting the corresponding revisions made in the updated version of the manuscript.

I. Reviewer 1

Conditional comments:

Comments 1: P2, Left A paragraph of "In addition, ...Wen et al., 2020)"I couldn't correctly understand the meaning of this paragraph, including the structure of the sentence. Please clarify the structure and add an explanation.

Response 1: Thank you for pointing out the confusion in the paragraph on page 2. We understand that the structure and flow of the sentence may not have been clear, and we appreciate your feedback. For clarity, we have restructured the sentence and added further explanation to provide a better understanding of the key concepts. The main purpose of this section is to discuss the existing post-processing methods and their limitations. It first discusses traditional post-processing methods, followed by post-processing methods that use external information, and finally discusses their limitations. The revised version can be found at P2, Left.

Original paragraph (for context):

"In addition, postprocessing technologies, such as the Gaussian–Newton method (Mu et al., 2018), multifrequency combination processing (Zhongchen Guo, 2023), and estimation processing in precise point positioning (PPP) (Zhao et al., 2023), are based mainly on a physical or empirical model to mitigate multipath interference. terrain assistance, net-work-based multipath mitigation (Klimenko et al., 2021), stochastic state estimation (Z. Hu et al., 2024), and sparsity-promoting regularization (Chen et al., 2019); however, these methods require external assistance or static long-term observation (Wen et al., 2019; Wen et al., 2020)."

Revised version:

"In addition, various postprocessing methods have been developed to mitigate multipath errors. Traditional post-processing methods include the Gaussian–Newton method (Mu et al., 2018), multifrequency combination processing (Zhongchen Guo, 2023), and estimation processing in precise point positioning (PPP) (Zhao et al., 2023), which are primarily based on physical or empirical models. Recently, some new postprocessing methods that utilize external information have emerged, such as the terrain network-based assistance method (Klimenko et al., 2021), stochastic state estimation (Z. Hu et al., 2024), and sparsity-promoting regularization (Chen et al., 2019). Nevertheless, these methods typically require additional external assistance or rely on static, long-term observation data (Wen et al., 2019; Wen et al., 2020)."

Explanation of the Changes:

- (1) Improved readability: The original paragraph contained a long, complex sentence that makes it difficult to follow. The sentence is replaced by two separated sentences, separating the description of the postprocessing technologies and the discussion of their limitations. This makes it easier for readers to follow the logical flow of the ideas.

- (2) Clarified the relationships: The revised version makes it clearer that the methods listed are aimed at mitigating multipath interference, and that they rely on physical or empirical models. This helps readers understand the purpose and function of each method.
- (3) Clear explanation of limitations: The limitations of these methods (requiring external assistance or long-term observation) are now presented as a separate idea, making it easier to grasp the potential drawbacks of these technologies.
- (4) Consistent citation format: We standardized the citation format to ensure consistency and improve readability.

Comments 2: *P3, Left, Line 5, “choke antennas”. I think it should be “choke ring antennas”. If so, please correct it.*

Response 2: Thank you for your thoughtful comment and for pointing out the use of the term "choke antennas." We have carefully reviewed the manuscript and revised the phrase to use the correct terminology. Additionally, upon further review, we noticed that the word in the conclusion was not entirely accurate. We have made the necessary revisions to improve clarity and precision. Below are the updated versions of the text on page 3, left, and the conclusion:

Revised Manuscript:

- (1) *“Consequently, this approach not only enhances the precision of satellite navigation positioning but also has the potential to reduce antenna-related expenses. The necessity for choke ring antennas may be obviated, leading to a more cost-effective solution in eliminating multipath error.”*
- (2) *“More importantly, we aim to substitute the choke ring antenna with a micro-jitter antenna for low-cost, high-precision measurements.”*

Thank you again for bringing this to our attention.

Comments 3: *Right, Equation (3). This autocorrelation function is a standard case (for example, PRN01 etc. in GPS). In the case of GPS, there are some other cases, so please mention this issue and add an explanation of how it will be handled in this study. In addition, “o.w.” is not clear. Please explain it in other words.*

Response 3: Thank you for your valuable feedback and for pointing out the need for clarification regarding the autocorrelation function and the use of "o.w.". In response to your question, we have made the following corrections.

- (1) Clarification Regarding Autocorrelation Function:

We agree that the autocorrelation function discussed is a standard case, such as PRN01 in GPS, but we also acknowledge that there are other cases in GPS with different signal characteristics. However, we use the standard autocorrelation function as an example for analysis to provide a clear illustration of the origins of multipath errors in this study [R1] [R2]. Moreover, the proposed method focuses on the cancellation of the multipath signals. Therefore, this method is applicable whenever the autocorrelation function exhibits symmetry. Below is the updated section of the manuscript on P3, Right, Eq. (3):

“Eq. (3) is a standard case (such as PRN01 in GPS). It is important to note that in GPS, there are other PRN codes with different autocorrelation functions. However, we usually use the standard case as an example to provide an analysis framework (Yu et al., 2024), and the methods presented can be extended to other cases, provided the autocorrelation function is symmetric.”

[R1] Yu L, Gao Y, Lian J, Li F, Gao F, & Wang T. Improving GNSS-RTK multipath error extraction with an integrated CEEMDAN and STD-based PCA algorithm. GPS Solutions, 2024, Vol. 28(4).

[R2] Xie G, GPS Principles and Receiver Design Revised Edition. Beijing: Electronic Industry Press, 2022.

(2) Clarification of "o.w.":

$$R(\tau) = \begin{cases} -\frac{1}{T_c}|\tau| + 1, & 0 \leq |\tau| \leq T_c \\ 0, & \text{otherwise} \end{cases} \quad (3)$$

We recognize that "o.w." might be unclear to readers. We have revised the manuscript to replace "o.w." with "otherwise" for better readability, which can be found on P3, right, Eq. (3) of the revised manuscript.

Thank you again for your insightful comments.

Comments 4: P3, Right, Fig. 2. Please add a description of the Early/Late correlator spacing in this diagram. Also, since this diagram shows a special case where the position of either the Early or Late correlator coincides with the point where the rate of change changes, please add an explanation of what this special positional relationship means. Additionally, I think code phase error will be code range error. If so, please modify this.

Response 4: Thank you for your helpful comments and suggestions. We appreciate your careful review and the points raised regarding the Early/Late correlator spacing, the special positional relationship in the diagram, and the terminology used for "code phase error."

(1) Early/Late Correlator Spacing:

We have added a description of the Early/Late correlator spacing in the diagram as requested. We sincerely apologize for any confusion caused by our oversight. In fact, d represents the early-late correlator spacing, and T_c is the chip length, which is typically set to 1. This should now provide a clearer understanding of the spacing between the correlators in the context of the diagram. Below is the updated section of the manuscript:

Fig. 2 Relationship between relative phase and multipath error.

“Fig.2 illustrates the GNSS signal autocorrelation function in the two cases of different relative phases, where the multipath amplitude ratio a_1 is 0.5, the E-L spacing $d = 0.7$ chip and the multipath delay $\Delta\tau_1$ is 0.5 chip.”

The revision regarding the correlator spacing d on P3, Eq. (2):

$$D = R(\hat{\tau}_0 - \tau_0 + d / 2) - R(\hat{\tau}_0 - \tau_0 - d / 2) \quad (2)$$

“where $\hat{\tau}_0$ is the estimate of the direct signal time delay, d is the early-late correlator spacing,”

The revision regarding the chip length T_c on P2, Eq. (3):

$$R(\tau) = \begin{cases} -\frac{1}{T_c} |\tau| + 1, & 0 \leq |\tau| \leq T_c \\ 0, & \text{otherwise} \end{cases} \quad (3)$$

“where T_c is chip length, and it is assumed to be 1 in this study.”

(2) Special Positional Relationship:

We agree that the diagram shows a special case where the position of either the Early or Late correlator coincides with the point where the rate of change changes. We sincerely apologize for any confusion caused by our oversight. In fact, we chose this special case solely for the convenience of helping readers quickly understand the errors introduced by multipath signals, as the transition at this special point is relatively clear. This is essentially due to the early/late correlator spacing, which leads to variations of the early and late correlation values. However, the method proposed in this paper is applicable under any traditional correlator spacing, as can be verified through both the theoretical analysis and numerical experiments presented later in the paper. We have also provided an explanation of this in the manuscript on P3 right:

“The diagram illustrates a specific correlator spacing, which is used solely to demonstrate the impact of multipath signals. Any traditional correlator spacing is applicable to the method proposed in this paper.”

(3) Code Phase Error vs. Code Range Error:

We agree that "code phase error" should be referred to as "code range error." We have updated the manuscript accordingly to reflect this correction on P4 left and Fig. 2:

“As a result, the code range error depicted in Fig. 2 will manifest as pseudo-random noise, which can be eliminated by averaging over a short period. We clarify the geometry relationship between antenna micro-jitter and multipath error through Fig. 3”

Thank you again for your insightful comments.

Comments 5: P3, Right, Line 17, An equation of e_{max} . Please add an explanation for variable d .

Response 5: Thank you for your helpful comments and suggestions. We sincerely apologize for any confusion caused by our oversight. In fact, d represents the early-late correlator spacing, and T_c represents the chip length, which is typically set to 1. We have provided a detailed explanation of d in Eq. (2) on P3 right:

$$D = R(\hat{\tau}_0 - \tau_0 + d / 2) - R(\hat{\tau}_0 - \tau_0 - d / 2)$$

“where $\hat{\tau}_0$ is the estimate of the direct signal time delay, d is the early-late correlator spacing.”

Thank you again for your insightful comments.

Comments 6. P3, Right, Line 28, “the carrier phase estimation error”. It is unclear what the carrier phase estimation error means, so please add some explanations.

Response 6: Thank you for your valuable feedback. We understand that the term "carrier phase estimation error" may not be fully clear to all readers. Upon receiving the navigation signal, the receiver must perform carrier stripping first, which involves estimating both the carrier frequency and carrier phase. The errors associated with these two processes can be represented as carrier phase errors. For clarity, we have added a more detailed explanation of what this term means in Eq. (4) on P4 and provided references [R1] that offer a more detailed explanation:

$$\varepsilon_{\max} = \frac{d}{2} \max \left[\sum_{i=1}^M a_i \cos(\theta_e + \Delta\theta_i) \right], \quad (4)$$

“where $\theta_e = \theta_0 - \hat{\theta}_0$ represents the difference between the carrier phase of the direct signal θ_0 and the local carrier phase estimate $\hat{\theta}_0$, i.e. the carrier phase estimation error.”

[R1] S. Braasch and A. J. van Dierendonck, "GPS receiver architectures and measurements," Proceedings of the IEEE, vol. 87, no. 1, pp. 48-64, Jan. 1999, doi: 10.1109/5.736341.

We hope this clarification helps the readers understand the term easily of the term. Thank you again for your helpful suggestion.

Comments 7: P3, Right, Line 31, “ T_c ”. It is unclear because T_c is not described in Equation (4). Please modify it correctly.

Response 7: We sincerely apologize for any confusion caused by our oversight. In fact, d represents the early-late correlator spacing, and T_c is the chip length, which is typically set as 1. We have revised the manuscript to clarify the confusion between d and T_c .

The revision regarding the correlator spacing d is made in Eq. (3) on P3 right:

$$D = R(\hat{\tau}_0 - \tau_0 + d/2) - R(\hat{\tau}_0 - \tau_0 - d/2) \quad (2)$$

“where $\hat{\tau}_0$ is the estimate of the direct signal time delay, d is the spacing between the late and early correlator”

The revision regarding the chip length T_c is made in Eq. (3) on P3 right:

$$R(\tau) = \begin{cases} -\frac{1}{T_c} |\tau| + 1, & 0 \leq |\tau| \leq T_c \\ 0, & \text{otherwise} \end{cases} \quad (3)$$

“where T_c is chip length, and it is assumed to be 1.”

We hope this clarification provides a better understanding of the term. Thank you again for your helpful suggestion.

Comments 8: P4, Left, Equation (5) and (6). Please mention in the text that this method may not make changes in path difference between a direct and a reflected signal when Psi is 90 degrees (parallel to the reflecting surface). Additionally, please add some explanations on c and f_c in the text.

Response 8: Thank you for your insightful comments. We have addressed the points you raised as follows:

(1) Clarification Regarding Path Differences at Psi is 90 degrees

We agree that this method may not make changes in the path difference between a direct and a reflected signal when Psi is 90 degrees (parallel to the reflecting surface). We Following your suggestion, we have added the text on P4 left Eq. (5) and (6):

“Note that this method may not make changes in path difference between a direct and a reflected signal when $\psi_n = 90^\circ$ or 270° (parallel to the reflecting surface).”

(2) Explanation of c and f_c :

We have included additional explanations of the parameters c (the speed of light) and f_c (the carrier frequency) in the manuscript for clarity on P4 Eq. (5) and (6):

“ c represents the speed of light in a vacuum.”

The explanation of f_c can be found in P3 Eq. (1):

“where $s(t)$ denotes the GNSS spreading code sequence. A , f_c , τ_0 , and θ_0 are amplitude,

carrier frequency, time delay and carrier phase of the direct signal, respectively.”

Thank you again for your constructive feedback. These additions should help clarify these aspects of the study.

Comments 9: P4, Right, Line 4, “Eq. (10)”. Although the equation numbers (10) are assigned, it should be better to follow the order in which they appear in the text. Also, please add some explanations of what τ_p and τ_q mean in this method.

Response 9: Thank you for your constructive feedback. We appreciate your suggestion regarding the equation numbering. In response, we have rearranged the equation numbers to follow the order in which they appear in the text, ensuring consistency and clarity. As a result, Eq. (10) has been renumbered as Eq. (7) in the revised manuscript on P4, right.

Additionally, Thank you for your reminder. τ_p and τ_q indeed play a significant role in this paper. They not only represent the threshold points of the variations in multipath errors but also influence how the complete mitigation of multipath errors is achieved in the subsequent sections. Therefore, we have added an explanation of the terms τ_p and τ_q as they are used in this method on P4, Right, Eq. (7):

“where $\tau_p = [1 + a_1 \cos(\theta_e(r, \psi) + \Delta\theta_1(r, \psi))]d/2$, $\tau_q = T_c + a_1 d \cos(\theta_e(r, \psi) + \Delta\theta_1(r, \psi))/2$. τ_p and τ_q represent the threshold points at which the $\Delta\tau_1(r_n, \psi_n)$ affects the $\varepsilon(a_1, \Delta\tau_1, \Delta\theta_1, r_n, \psi_n)$, primarily influenced by d . $\varepsilon(a_1, \Delta\tau_1, \Delta\theta_1, r_n, \psi_n)$ will exhibit a uniform distribution with a mean of zero when $\tau_p < \Delta\tau_1(r_n, \psi_n) < \tau_q$ for different $\Delta\theta_1$, which can be completely compensated with the method proposed in this paper.”

Thank you again for your valuable comments. We believe these revisions improve the clarity and flow of the manuscript.

Comments 10: P4, Right, Line 28, “Fig. 4” I could not find Fig 4, so please check and modify it.

Response 10: Thank you for your careful review. We apologize for the oversight regarding the reference to "Fig. 4." After checking, we found a formatting error in this section. The reference to "Fig. 4" should refer to "Fig. 5" of the original manuscript, and this paragraph should appear at the end of Section 2.2. We have corrected this. The revised version of Fig. 4 can now be found on the right side of P5:

Fig. 4 Bias of Direct signal estimation.

“It can be observed in Fig. 4 that bias is fully negated upon averaging with symmetric jitter. Conversely, the bias post-averaging remains unmitigated with asymmetric jitter. Thus, adjustments based on the jitter trajectory are required when symmetric jitter cannot be implemented, yet achieving precise corrections is notably difficult. On the other hand, the variance of the direct signal bias decreases progressively as the number of samples increases. Once the number of samples exceeds 1000, the variance reduction exhibits a linear trend and below 10^{-5} m, meeting the requirements for real-time high-precision positioning.”

Thank you again for your valuable comments.

Comments 11: P4, Right, Line 42, “10-510-5 meters” Please modify it correctly.

Response 11: Thank you for your careful review. After checking, we found a formatting error in this section. This paragraph “The Fig. 4... positioning requirements.” on P4 of the original manuscript should appear at the end of Section 2.2. We have corrected this. The revised version of this paragraph can now be found on the right side of P5:

“It can be observed in Fig. 4 that bias is fully negated upon averaging with symmetric jitter. Conversely, the bias post-averaging remains unmitigated with asymmetric jitter. Thus, adjustments based on the jitter trajectory are required when symmetric jitter cannot be implemented, yet achieving precise corrections is notably difficult. On the other hand, the variance of the direct signal bias decreases progressively as the number of samples increases. Once the number of samples exceeds 1000, the variance reduction exhibits a linear trend and below 10^{-5} m, meeting the requirements for real-time high-precision positioning.”

Thank you again for your valuable comments.

Comments 12: P5, Right, Equation (13). Please add some explanations on N and the transformation of the variance formula.

Response 12: Thank you for your valuable feedback. We apologize for the confusion caused by our mistake. The distribution in Eq. (12) in the original manuscript was incorrect, and we have corrected it to $r_n \sin(\beta + \psi_n + \alpha) \sim U(\delta_-, \delta_+) \sim U(\delta_-, \delta_+)$ and $\delta = \frac{1}{N} \sum_{n=1}^N r_n \sin(\beta + \psi_n + \alpha)$. Then, the following process can be derived using the variance formula of the uniform distribution, which can be found on P5, left, Eq. (14). It is worth noting that Eq. (13) in the original manuscript is renumbered as Eq. (14) in the revised version.

$$D(\delta) = \frac{1}{N^2} D\left[\sum_{n=1}^N r_n \sin(\beta + \psi_n + \alpha)\right] = \frac{(\delta_+ - \delta_-)^2}{12N} \quad (14)$$

Thank you again for your valuable comments.

Comments 13: P6, Left, Line 45, A sentence of “However, ... error.”. Please add information on evidence from the previous research articles etc.

Response 13: Thank you for your suggestion to add supporting evidence from previous research in the sentence on Page 6, Line 45. We agree that referencing prior studies would strengthen the argument and provide additional context to our findings.

The latest research on antenna movement to counter multipath errors comes from a scholar in

Japan, who uses a rotating antenna to suppress multipath errors. This method reduces the positioning error from 18.96 m to 2.83 m. However, the study does not explore the underlying mechanisms of how antenna movement mitigates multipath errors in depth. Additionally, the rotating antenna has a large radius of 25 cm, which limits its potential application scenarios.

In contrast, we project any antenna movement method onto linear jitter and delve deeper into the mechanisms by which antenna movement suppresses multipath errors. The basic principle of antenna motion for counteracting multipath errors is the antenna's movement perpendicular to the reflective surfaces. Although a rotating antenna may seem to offer superior performance due to its consideration of reflections from all directions, linear jitter can similarly address reflections from all directions when reflective surfaces are randomly distributed. Furthermore, if there exists a dominant reflective surface, a linear jitter can more effectively target it. A detailed analysis of this issue is provided in Section 3.2 of the revised manuscript. Both our simulations and practical experiments have confirmed this. There is no significant difference in error suppression between linear and rotating antennas, but linear jitter saves considerable physical space, expanding the applicability of this method. In conclusion, linear jitter demonstrates greater potential for practical application.

The relevant information has also been added to the manuscript, which can be found in Remark 1 on page 6 of the revised version:

“In contrast, a rotating antenna appears to offer a superior effect on multipath signals when multiple distinct reflective surfaces are present in theory, as it can account for reflections from all directions (Suzuki et al., 2020). However, subsequent experiments demonstrate that linear jitter can effectively eliminate errors with almost the same performance compared to a rotating antenna, as the randomly distributed reflective surfaces in the real world can counteract the effect of jitter direction.”

Comments 14: P7, Left, Fig. 7. Please mention in the text that when jitter angles are $1/2 * \pi$, $3/2 * \pi$, the jitter directions are parallel to the reflecting surface and it causes diverges. Also, I believe that the actual minimum jitter amplitude needs to be determined by comparing it with the random error component in the pseudorange measurements, so please add some explanations on this issue in the text.

Response 14: Thank you for your valuable feedback. We appreciate your insightful suggestions regarding the interpretation of jitter angles and the discussion of the minimum jitter amplitude.

(1) Jitter Angles and Divergence

We agree with your observation that when jitter angles are $1/2 * \pi$, $3/2 * \pi$, the jitter directions are parallel to the reflecting surface and it causes diverges. This effect is indeed crucial for understanding the behavior of the jitter in our system. We have added a clarification in the revised manuscript, particularly in Fig. 6 on the right side of P7 (which is Fig. 7 in the original manuscript). The revised text is as follows:

“It can be seen from the comparison in Fig. 6 that the multipath error decreases different values with the various jitter amplitude and angle, where the jitter angle and amplitude of minimum error are strongly consistent with the theoretical analysis. The above jitter angle and amplitude are equivalent to the antenna jittering $\pm 5\text{cm}$ along the direction normal to the reflecting surface, such a jitter amplitude is acceptable in the real environment. Additionally, it should be noted that multipath error diverges when jitter angles are $\pi / 2$ and $3\pi / 2$, i.e., the jitter directions are parallel

to the reflecting surface. This also confirms the derivation in Theorem 1, which emphasizes the need to select a reasonable angle.”

Fig. 6 The variation trend of multipath error versus jitter amplitude and angle.

(2) Actual minimum jitter amplitude

Regarding your second point about the determination of the minimum jitter amplitude, we agree that it is essential to compare the jitter amplitude with the random error component inherent in pseudorange measurements in practical application. This comparison provides a more accurate assessment of the jittering impact on the overall measurement accuracy.

However, this paper primarily investigates the impact of antenna motion on mitigating multipath errors. Therefore, we have temporarily disregarded the random errors in the estimation process to clearly illustrate the effects of antenna motion [R3] [R4]. The random errors in pseudorange estimation are related to various factors, including thermal noise, dynamic stress, and Allan variance [R5]. The analysis of these random errors is an important and relatively independent topic, which will be progressively addressed in our future research. To clarify this issue, we have expanded the explanation in the text P6 left as follows:

“Moreover, it is important to emphasize that the actual minimum jitter amplitude needs to be determined by comparing it with the random error component in the pseudorange measurements. However, this paper aims to transform multipath errors into random errors and explore the impact of antenna jitter patterns on error transformation. Therefore, random errors of the pseudorange have been temporarily neglected to demonstrate the impact of antenna jitter. A detailed discussion of random errors will be discussed in our future work.”

[R3] Song W, Wu Q, Gong X, Zheng F, and Lou Y. Corrections of BDS Code Multipath Error in Geostationary Orbit Satellite and Their Application in Precise Data Processing. Sensors (Basel), 2019, Vol. 19(12): 2737.

[R4] Zhou H, Wang X, Zhong S, Li Y, and Xi K. Multipath error extraction and mitigation based on refined wavelet level and threshold selection. GPS Solutions, 2024, Vol. 28(4).

[R5] Xie G, GPS Principles and Receiver Design Revised Edition. Beijing: Electronic Industry Press, 2022.

Thank you again for your constructive suggestions. We hope these revisions meet your expectations and look forward to any further feedback.

Comments 15: P8, Left, Fig. 9 The figure contains the words "Sim" and "Ana". Please add some explanations to them, including the differences between them. The same would be for Fig10.

Response 15: Thank you for your valuable suggestion regarding the need for clarification of the terms "Sim" and "Ana" in Fig. 9 and Fig. 10. We agree that providing an explanation of these terms

and highlighting the differences between them will improve the readability of the manuscript. Specifically, "Sim" refers to the simulated results obtained through numerical simulations based on the proposed model and parameters. "Ana" refers to the analytical results, which are derived from the theoretical analysis or closed-form expressions presented in the paper. The comparison between the "Sim" and "Ana" under the same conditions can visually verify the correctness of our theoretical analysis. We have added the supplementary information in the revised manuscript, which can be found in Fig. 7 on page 8:

"Fig. 7 illustrates the variation trend of multipath error versus jitter amplitude for different time delays, where "Sim" refers to the simulated results obtained through numerical simulations, while "Ana" refers to the analytical results, which are derived from the theoretical analysis in the paper."

Thank you again for your helpful suggestions. We believe this addition will provide a clearer understanding of the results presented in these figures.

Comments 16: *P8, Right, Line 5, A paragraph of "In Fig. 10, ... the antenna jitter.". Actually, it is needed to take into account random errors in the pseudorange measurements. Please mention it in this paragraph.*

Response 16: Thank you for your valuable feedback. We agree that random errors in pseudorange measurements should be considered, which has led to inaccuracy in our statement in this paragraph, where we previously indicated that the multipath error can be reduced to zero. This has now been corrected. However, the core focus of this paper is to transform multipath errors into random errors, and thus we emphasize the impact of antenna jitter on the conversion of multipath errors. Moreover, the suppression techniques for random errors are already relatively well-established. Therefore, to more intuitively demonstrate the impact of antenna jitter, other random errors in the pseudorange measurements are temporarily neglected [R6] [R7].

Nevertheless, random errors do affect the antenna jitter discussed in this paper. The random errors in pseudorange estimation are related to various factors, including Gaussian white noise, dynamic stress, and Allan variance, with the impact of Gaussian white noise being the most significant [3]. Therefore, we have considered the impact of Gaussian white noise and the multipath error suppression performance under different signal-to-noise ratios (SNR) is illustrated in Fig. 12. Finally, the explanation of the random errors in pseudorange measurements is provided in the revised manuscript, which can be found in Fig. 9 on page 8 and Fig. 12 on page 10:

“Fig. 12 Comparison of Multipath Mitigation Algorithms

As shown in Fig. 12, it is evident that the antenna jitter possesses distinct advantages in terms

of robustness and efficacy. This technique demonstrates remarkable performance under low SNR conditions, achieving steady-state convergence 10-15 dB earlier than other methods, which is particularly suited to counteract the complexities of urban and other challenging environments. Furthermore, the error during the steady state of this method is reduced by approximately 80%-90% compared to traditional narrow correlation and Early-Prompt-Late (EPL) methods, even surpassing the performance of more complex algorithms such as Code Correlation Reference Waveform (CCRW) (C. Xu et al., 2016) and Double-delta, which is attributed to the introduction of new observables through motion. Thus, this method enables the direct application to conventional receivers as an independent parameter. This results in low-complexity, high-precision positioning and it can be integrated with other high-precision algorithms to meet the ultra-high-precision requirements of monitoring stations.”

“Fig. 9 Multipath error envelopes with different correlation distances.

In Fig. 9, there is a clear downward trend of multipath when combined with narrow correlation technology. In particular, the envelope peak value of the correlation distance of 1/8 chip can be further reduced by 4/5, and the effective length of the envelope can be reduced by 7/8 compared to the correlation distance of 1 chip. As a result, the envelope peak value can be reduced by 98%, equivalent to around 0.005 chips. Moreover, approximately 87.5% of the overall time delay of the multipath error can be minimized to approach zero when random errors in the pseudorange measurements are not considered, indicating the theoretical effectiveness of the antenna jitter.”

Thank you again for your helpful suggestions.

[R6] Song W, Wu Q, Gong X, Zheng F, and Lou Y. Corrections of BDS Code Multipath Error in Geostationary Orbit Satellite and Their Application in Precise Data Processing. Sensors (Basel), 2019, Vol. 19(12): 2737.

[R7] Zhou H, Wang X, Zhong S, Li Y, and Xi K. Multipath error extraction and mitigation based on refined wavelet level and threshold selection. GPS Solutions,2024, Vol. 28(4).

Comments 17: P10. Left, Fig. 13. Normally, a signal strength of more than about 26 dB in C/N0 is required to capture GNSS signals. Please add some explanation on why signal tracking is possible even at 0 dB, including the correlator spacing set in this figure.

Response 17: Thank you for your valuable comment. We apologize for the confusion caused by our oversight. The correct unit for the 26 dB you mentioned should be dBHz, which is the unit for carrier-to-noise density ratio (C/N₀). However, in this study, we used signal-to-noise ratio (SNR) as the horizontal axis for comparison, with the unit being dB. The difference between the two is the receiver bandwidth B_e and coherent integration time T_{coh} .

$$SNR = C / N_0 - 10 \lg(B_e T_{coh}) \quad (26)$$

where the unit of C / N_0 is dBHz, the unit of SNR is dB, the unit of T_{coh} is s, and the unit of B_e is Hz. In this simulation experiment, a receiver bandwidth was set $B_e = 1\text{MHz}$ and coherent integration time was set $T_{coh} = 1\text{ms}$. Therefore, if the horizontal axis in the figure were converted to C/N_0 , it would correspond to 30-80dBHz. We regret that we did not specify the bandwidth and T_{coh} used in the simulation experiment, and a more detailed conversion can be found in [R1]. In response to your suggestion, we have revised the manuscript to include a more detailed explanation of the signal-tracking performance at shallow C/N_0 levels, and we have added a description of the correlator spacing, which can be found on P10, left, Eq. (26).

“The amplitude of the antenna jitter was determined based on Eq. (16) with an initial bias set at 0.2 chips, and the correlator spacing $d = 1/8$ chip. Considering the influence of random errors, we use the signal-to-noise ratio (SNR) as the variable to observe the error suppression effect. The relationship between SNR and carrier-to-noise density ratio (C / N_0) is (Braasch et al., 1999):

$$SNR = C / N_0 - 10 \lg(B_e T_{coh}) \quad (26)$$

where the unit of C/N_0 is dBHz, the unit of SNR is dB, the receiver bandwidth B_e was set to $B_e = 1\text{MHz}$, and the unit of coherent time T_{coh} was set to $T_{coh} = 1\text{ms}$.”

[R1] S. Braasch and A. J. van Dierendonck, "GPS receiver architectures and measurements," in Proceedings of the IEEE, vol. 87, no. 1, pp. 48-64, Jan. 1999, doi: 10.1109/5.736341.

Thank you again for your helpful suggestions.

Comments 18: *P10, Left, Line 13, “as shown in Figure 12” I think it should be “as shown in Figure 13”*

Response 18: We appreciate your attention to this matter and sincerely apologize for the formatting issue. Upon further inspection, we identified that the problem stemmed from an error in the figure numbering in the original manuscript (Figure 4 was omitted). This has now been corrected, and in the revised version, the figure has been renumbered as Fig 12 on P10 left.

Fig. 12 Comparison of Multipath Mitigation Algorithms

“It is evident that the antenna jitter possesses distinct advantages in terms of robustness and efficacy through Fig. 12.”

Thank you again for your helpful suggestions.

Comments 19: *P10, Left, Line 23, “CCRW”. Please add description of the original words.*

Response 19: Thank you very much for your thoughtful comments and valuable suggestions. We

sincerely apologize for the oversight on our part. The Code Correlation Reference Waveform (CCRW) is used in the receiver to correlate the incoming satellite signals with a known reference code, such as the P-code or C/A code used in GPS. This correlation helps the receiver determine the signal's time-of-arrival (TOA) and, consequently, the user's position. The reference waveform is crucial for mitigating errors caused by multipath, noise, and signal degradation [R8]. The following are our revisions to the manuscript, which you can find on P10 left.

“Furthermore, the error during the steady state of this method is reduced by approximately 80%-90% compared to traditional narrow correlation and Early-Prompt-Late (EPL) methods, even surpassing the performance of more complex algorithms such as Code Correlation Reference Waveform (CCRW) (C. Xu et al., 2016) and Double-delta, which is attributed to the introduction of new observables through motion.”

[R8] Xu CT, Liu Z, Tang XM, Wang FX. A Design Method of Code Correlation Reference Waveform in GNSS Based on Least-Squares Fitting. SENSORS, 2016, Vol. 16(8): 1194.

Comments 20: P10, Right, Line 32, “In Fig. 14, No. 1 and No. 59 satellites” Please add the core satellite name of them, i.e. Beidou.

Response 20: Thank you for your helpful comment. We appreciate your suggestion to provide more specific information regarding the satellite names. In response, we have updated the manuscript to clarify that the satellites referred to as "No. 1" and "No. 59" in Fig. 13 are part of the Beidou satellite constellation, which can be found on P10 right in the revised manuscript:

"In Fig. 13, Beidou No. 1 and No. 59 satellites have azimuths of 127° and 133°, respectively, where the main reflective surface is the wall in the 300° direction. "

We believe this addition enhances the clarity of the figure and ensures the information is more comprehensive for readers. Thank you again for pointing this out.

Comments 21: P21, Left, Table 4. Please add product information on GNSS antenna and receiver in the Table or in the text.

Response 21: Thank you for your valuable suggestion. We agree that providing additional details about the GNSS antenna and receiver will improve the clarity and completeness of the manuscript. The monitoring antenna and receiver used in this experiment were both independently developed by our institution. Their parameter requirements are fully customizable and under our control; however, they are not currently available for commercial sale. To provide better clarity, we have included a detailed description of the monitoring antenna and receiver at the end of the manuscript.

“Appendix A

This appendix provides the detailed specifications of the monitoring antenna and receiver used in the experiment. Both components were independently developed by our institution, as illustrated in Fig. 20:

Fig. 20 Monitoring receiver and antenna.

The performance parameters related to this experiment are as follows:

Table 11 Monitoring antenna parameters:

Parameters	Values
Gain	$\geq 40\text{dB}$
Support system	BD/GPS/GLONASS/Galileo
Polarization method	Right-circular polarization
Choke ring	Yes
Voltage	DC 5V
Impedance	$50\ \Omega$

Table 12 Monitoring receiver parameters:

Parameters	Values
Pseudorange accuracy	10 cm
BDS	B1, B3
GPS	L1, L2
GLONASS	L1, L2
Galileo	E1 E5b
Time to first fix	Hot(<15s) Cold(<60s)''

Thank you again for pointing this out.

Comments 22: P14, Left, Table 10 Please add product information on GNSS antennas in the Table or in the text.

Response 22: Thank you for your constructive feedback. We agree that adding product information about the GNSS antennas will improve the clarity of the manuscript. We have added information on the GNSS monitoring antenna in Appendix A. Additionally, for comparison, we have included the information on the vehicle-mounted antennas used for comparison in Appendix B. We hope this will be helpful to you:

“Appendix B

This appendix provides information on commonly available commercial vehicle-mounted antennas selected for comparison in this experiment. A visual comparison between these antennas and the monitoring antenna is presented in Fig. 21. The performance parameters related to this experiment are listed in Table 13:”

Fig. 21 Monitoring and vehicle antenna.

Table 13 Vehicle antenna parameters:

Parameters	Values
Gain	$28\pm 2\ \text{dB}$
Support system	BD/GPS
Polarization method	Right-circular polarization
Choke ring	NO
Voltage	DC 3-5V
Impedance	$50\ \Omega$ ”

Thank you again for pointing this out.

Comments 23: *P14, Left, Line 12, A sentence of “This divergence is attributed to the antenna gain, which leads to a reasonable inference.” One possibility could be remained that the antenna phase characteristic is not uniform in the vehicle antenna. Therefore, please modify this sentence as “This divergence is attributed to the antenna gain, which could lead to a reasonable inference.” etc.*

Response 23: Thank you for your thoughtful comment. We agree that it is important to acknowledge the possibility of non-uniform antenna phase characteristics in the vehicle antenna. In response to your suggestion, we have revised the sentence, which can be found above the Conclusion on page 14 of the revised manuscript.

“This divergence is attributed to the antenna gain, which could lead to a reasonable inference. Therefore, if the CNR were equivalent, the ordinary antenna could fully supplant the costly and complex antenna when subjected to the proposed method.”

We believe this modification better captures the nuances of the situation. Thank you again for your valuable input.

Other comments:

Comments A1: *P12, Left, Line 5, “Specifically, the B1”-> Specifically, MS error for the B1.*

Response A1: Thank you for your suggestion. We have made the recommended change to improve clarity. The sentence has been updated, which can be found in Fig.15 on page 12 of the revised manuscript.

“Specifically, MS error for the B1 carrier frequency for the No.1 satellite, decreased from 0.33m to 0.15m post antenna jitter.”

We believe this revision enhances the accuracy of the statement. Thank you again for your valuable input.

Comments A2: *P13, Left, Line 21, “where we choose the root ... 60 satellites.” This part would be not necessary.*

Response A2: Thank you for your helpful feedback. We agree that this part may not be necessary for the clarity of the manuscript. As a result, we have removed the sentence. We believe this simplification improves the readability of the text. Thank you again for your suggestion.

Comments A3: *P13, Right, Line 9, “Table 6”. It would be removed.*

Response A3: Thank you for your suggestion. We have removed the reference to "Table 6" as recommended. We believe this revision improves the flow of the text.

II. Reviewer 2

Comments 1: *As mentioned in the abstract, The results show that an antenna jitter of $\pm 6.2\text{cm}$ can reduce the pseudo-range mean square error by 72.42% and the 3D localization error by 62.31% compared with a stationary antenna. This experimental result is only for a specific experimental situation and does not have universal reference significance, so it is not suitable to appear in the abstract.*

Response 1: Thank you for your valuable feedback. We agree with your observation regarding the experimental result presented in the abstract. The result concerning the antenna jitter and its impact on the pseudorange mean square error and 3D localization error is specific to the experimental conditions we used and may not be universally applicable. We appreciate your suggestion to revise the abstract to avoid presenting such specific results without sufficient generalization.

As a result, we have modified the abstract to remove this detailed experimental outcome and instead focused on the broader findings of the study. The updated abstract now emphasizes the overall contribution of the research and the methodology, rather than specific experimental results. The revised abstract reads as follows:

“Global Navigation Satellite System (GNSS) signals often encounter ... a broad range of receivers. This paper provides a quantitative analysis of the effect of jitter angle, amplitude, and environmental factors on the effectiveness of error cancellation, identifying the minimum jitter amplitude required for effective error reduction in straight-line jitter scenarios and performing experimental evaluations under various conditions. The results indicate that the proposed method significantly enhances robustness, efficiency, and applicability compared to traditional approaches. Furthermore, it holds promising potential to reduce the cost of high-precision antennas from thousands to merely a few tens of dollars, providing a substantial breakthrough in cost-effectiveness.”

We hope this revision addresses your concern, and we believe it now provides a more general overview of the study’s contributions. Thank you again for your constructive input.

Comments 2: *The manuscript describes the method of determining minimum jitter amplitude in a lot of lengths, but the process and results of determining minimum jitter amplitude are not reflected in the experimental part, and the basis for selecting an antenna jitter of $\pm 6.2\text{cm}$ is not clarified. This shows that there is a separation between the experimental process and the theoretical analysis.*

Response 2: We thank the reviewer for highlighting the need for clarification regarding the determination of the minimum jitter amplitude and the basis for selecting the antenna jitter of ± 6.2 cm. The theoretical analysis in this paper aims to explore the theoretical validity of antenna jitter and its influencing factors, thereby guiding the experimental measurements. It is undeniable that theory cannot fully encompass the complexities of real-world environments. Therefore, we have added some simulation experiments to approximate the real environment, validate the accuracy of the theoretical analysis, and further guide the experimental measurements. The specific responses are as follows:

(1) Justification for ± 6.2 cm Jitter Amplitude:

We apologize for not clearly explaining the origin of the 6.2 cm value in the manuscript. This value is derived from the experimental setup we selected and the result of Theorem 1. Specifically, Table 4 lists the actual environmental conditions, such as satellite elevation angles β and the angle of the reflecting surface α . In this scenario, the satellite elevation angles are 40.3° and 41.4° , the

reflection surface angle is 90° , and the satellite signal carrier frequency is 1561.098 MHz. Further, the minimum jitter amplitude for this scenario can be determined based on the conclusion from Theorem 1:

$$r_M = \frac{c}{4f_c |\sin(\alpha + \beta)\cos\psi_n|} \quad (16)$$

It is worth noting that the 6.2 cm is specific to the environment discussed in this paper. However, the proposed method is applicable across different environments. We have also made revisions to the manuscript, which can be found in Table 4 on page 11:

“The theoretical minimum jitter amplitude can be derived through Eq. (27) based on Table 4 and Theorem 1, where the carrier frequency $f_c = 1561.098\text{MHz}$ ”

$$r_M = \frac{c}{4f_c |\sin(\alpha + \beta)\cos\psi_n|} \quad (27)$$

$$\approx 6.2\text{cm}$$

(2) Clarification of the Determination of Minimum Jitter Amplitude:

We agree with your observation that there is a separation between the theoretical analysis and experimental results. We also encountered this issue during our research. We only considered the case where a single reflective surface exists in the environment, which allows for the precise determination of jitter angle and amplitude in the signal modeling and the process of determining the minimum jitter amplitude based on Theorem 1. However, such an ideal scenario does not exist in real-world conditions. To address this, we have added a section on simulation experiments in both the theoretical analysis and experimental verification to specifically consider multiple reflective surfaces, as shown in Section 3.2 of the revised manuscript.

In Section 3.2, we categorize real-world environments into two typical cases for analysis. The first case involves a primary reflective surface near the antenna, typically a building or wall. In this scenario, the primary reflective surface can be directly identified using Theorem 1, allowing for the determination of vertical jitter relative to the main reflective surface. Simulation results show that, compared to random jitter, applying the conclusion from Theorem 1 leads to a more significant improvement in error suppression. This analysis can be found in Figure 10 on P9 of the revised manuscript:

Fig. 10 The variation trend of multipath error versus the number of reflective surfaces under different antenna states.

“The comparison of Fig.10 reveals that the optimal jitter mode, previously determined for a single reflective surface, remains effective even when multiple reflective surfaces are present.”

Specifically, the MS error associated with the jitter antenna is markedly lower than the static antenna, and this discrepancy tends to increase with the addition of reflective surfaces. Furthermore, the optimal jitter method yields markedly smaller multipath errors than other antenna states, because it accounts for the calibration of the primary reflective surface. This suggests that the optimal jitter direction and angle play an important role in real scenarios if the primary reflective surface can be calibrated.”

The second case involves scenarios where there is no distinct primary reflective surface around the antenna, which is commonly observed in field monitoring stations. In this case, we apply Theorem 1 to arbitrarily select a reflective surface and compare the results with those of random direction and amplitude jitter. The results show that the proposed method performs similarly to random jitter in terms of direction and amplitude, highlighting the robustness of our approach. Moreover, the method enables the rapid determination of the minimum jitter amplitude, thus saving significant physical space. These findings are presented in Fig. 11 on P9 of the revised manuscript:

“Fig. 11 The variation trend of multipath error versus the number of reflective surfaces under no main reflective surface.

It can be observed that an increase in jitter amplitude primarily results in a reduction of variance within scenarios featuring a higher number of reflective surfaces in Fig. 11. Conversely, the mean value of the multipath error tends to increase, which underscores the efficacy of an optimally determined jitter amplitude. Furthermore, the discrepancy between the multipath errors generated by the optimal jitter method and those produced by a rotating antenna is found to be minimal. This finding validates that linear jitter is equally effective as the rotating antenna technique multiple reflective surfaces, with the added benefit of conserving space.”

(3) Inclusion of Experimental Verification:

To validate the theoretical results with real experiments, we conducted two sets of experiments. First, we verified the accuracy of the minimum jitter amplitude and direction as derived from Theorem 1 in an open environment. The results show that even jittering parallel to the primary reflective surface can provide some degree of error suppression in real-world environments. However, the jitter angle and amplitude derived from Theorem 1 yield the optimal suppression effect. This analysis is presented in Section 4.1 of the revised manuscript.

Secondly, we deliberately created a complex multipath environment to validate the effectiveness of linear jitter. The results show that, in more complex environments, linear jitter with the amplitude determined by Theorem 1 is highly effective at suppressing multipath errors. In this scenario, the suppression effect becomes less sensitive to the direction of jitter. More importantly, the suppression

performance in the complex environment is even more pronounced than in the open environment, further demonstrating the robustness and versatility of our method. This suggests that, in the future, our approach could potentially overcome the environmental constraints typically associated with monitoring antenna installation. The analysis of this part is presented in Section 4.2 of the revised manuscript.

We hope that the above response addresses your concerns. Once again, we would like to express our sincere gratitude for your valuable suggestions.

Comments 3: *There are typographical problems and writing errors in the article, for example,*

a) *We clarify the geometric relationship between the antenna jitter and multipath error through Fig. 3 The sentence lacks a period at the end.*

b) *There is a large amount of blank space in the page before formula (10) and formula (24).*

c) *There is a meaningless black block at the top of Figure 7.*

d) *Multiple tables in the article are disconnected, affecting reading.*

e) *The antenna parameters are delineated in the Table 9Table 6. E) The antenna parameters are delineated in the Table 9Table 6. The absence of and in the sentence above.*

Response 3: Thank you for your careful review and for pointing out the typographical issues. We appreciate your attention to detail and have made the necessary corrections to address each of the points raised:

a) Issue with Period in Sentence

We apologize for the oversight. We have added the missing period to the sentence. The revised sentence can be found on P4 Fig3:

"We clarify the geometric relationship between the antenna jitter and multipath error through Fig. 3."

b) Blank Space Before Formula (10) and Formula (24)

Thank you for pointing this out. We have adjusted the formatting to reduce the excessive blank space on the page before formulas (10) and (24), ensuring better alignment and flow of content. In addition, we have also reorganized the formatting and ordering of the equations. Specifically, Eq. (10) and (24) in the original manuscript have been reordered as Eq. (7) and (24) in the revised version.

$$\begin{aligned}
 \varepsilon(a_i, \Delta\tau_i, \Delta\theta_i, r_n, \psi_n) = & \begin{cases} \frac{\Delta\tau_1(r_n, \psi_n) a_i \cos(\theta_e + \Delta\theta_1(r_n, \psi_n))}{1 + a_i \cos(\theta_e + \Delta\theta_1(r_n, \psi_n))}, 0 < \Delta \\ \frac{a_i d \cos(\theta_e + \Delta\theta_1(r_n, \psi_n))}{2a_0}, \tau_p < \Delta\tau_1(r_n, \psi_n) \\ \frac{a_i \cos(\theta_e + \Delta\theta_1(r_n, \psi_n))}{2 - a_i \cos(\theta_e + \Delta\theta_1(r_n, \psi_n))} \\ \times \left(T_c + \frac{d}{2} - \Delta\tau_1(r_n, \psi_n) \right), \tau_q < \Delta\tau_1(r_n, \psi_n) \\ 0, \Delta\tau_1(r_n, \psi_n) > T_c + \frac{d}{2}, \end{cases} & (7) \\
 \varepsilon(a_i, \Delta\tau_i, \Delta\theta_i, r_n, \psi_n) = & \begin{cases} \sum_{i=1}^M \frac{\Delta\tau_i(r_n, \psi_n) a_i \cos(\theta_e + \Delta\theta_i(r, \psi))}{1 + a_i \cos(\theta_e + \Delta\theta_i(r, \psi))}, 0 < \Delta\tau_i(r, \\ \sum_{i=1}^M \frac{a_i d \cos(\theta_e + \Delta\theta_i(r_n, \psi_n))}{2a_0}, \tau_p < \Delta\tau_i(r, \psi) \leq \tau \\ \sum_{i=1}^M \frac{a_i \cos(\theta_e + \Delta\theta_i(r_n, \psi_n))}{2 - a_i \cos(\theta_e + \Delta\theta_i(r_n, \psi_n))} \\ \times \left(T_c + \frac{d}{2} - \Delta\tau_i(r_n, \psi_n) \right), \tau_q < \Delta\tau_i(r_n, \psi_n) \leq T_c + \\ 0, \Delta\tau_i(r_n, \psi_n) > T_c + \frac{d}{2} \end{cases} & (24)
 \end{aligned}$$

c) Black Block in Figure 7

"There is a meaningless black block at the top of Figure 7."

We apologize for this issue. The black block was inadvertently included during the figure formatting process. We have corrected the figure and removed the black block, ensuring that Figure 7 is clear and properly formatted now. It is worth mentioning that, Fig 7 in the original manuscript has been reordered as Fig 6 in the revised version on P7 right.

d) Disconnected Tables Affecting Readability

We have reviewed the layout of the tables and have corrected any formatting issues to ensure that the tables are properly aligned and consistent throughout the manuscript. The disconnected tables have been fixed to improve readability and presentation. For example, Table 7 and Table 8, as well as Table 9 and Table 10, have been properly aligned and formatted to ensure consistency and readability.

e) Incorrect Reference to Table 9 and Table 6

"The antenna parameters are delineated in the Table 9Table 6. The absence of 'and' in the sentence above."

Thank you for catching this error. We sincerely apologize for the oversight. There was an issue with the citation in that section. In fact, only Table 9 provides the antenna parameter comparison. We have corrected the alignment, and you can now find the revised information in Section 4.4 of the revised manuscript on P13.

"The antenna parameters are delineated in Table 9"

We hope these revisions address your concerns, and we believe they have significantly improved the clarity and formatting of the manuscript.

Comments 4: *There are problems in the drawing quality of the pictures in the article Fig.7, Fig.19, Fig.20 images are blurred, and it is suggested to redraw Fig.14 and Fig.17.*

Response 4: Thank you for your valuable feedback regarding the quality of the images in the manuscript. We apologize for the issues related to the clarity of Figures 7, 19, and 20. We understand that blurred images can impact the overall quality and readability of the manuscript. The images with improved clarity are displayed as follows. It should be noted that Figures 7, 19, and 20 in the original manuscript have been reordered as Figures 6, 18, and 19 in the revised version. We sincerely apologize for any confusion this may have caused during your review.

Fig. 6 The variation trend of multipath error versus jitter amplitude and angle.

Fig. 19 3D positioning error in different antennas.

(a) Open platform

(b) Complex platform

Fig. 18 3D positioning error in different platforms.

In addition, we have redrawn Fig. 14 and 17, including changes to some components, adjustments to color schemes, and modifications to the layout. It is important to note that Fig. 14 and 17 in the original manuscript have been reordered as Fig. 13 and 16 in the revised version. The specific revisions are shown in the figures below. If you have any further suggestions, we would greatly appreciate your feedback.

Fig. 13 Measured scene in the open platform.

Fig. 16 Placement of the artificially complex platform.”

We believe these changes significantly enhance the visual quality of the figures and hope they are now clear and well-presented.

Thank you again for your constructive comments.

Responses to the Reviewers

Manuscript ID: COMMS-24-0484A

Title: A Multipath Error Cancellation Method Based on Antenna Jitter

The authors would like to thank the reviewers for volunteering their time to review our paper and provide us with valuable comments. We have revised our manuscript based on the comments and suggestions. The following are our point-by-point responses to the raised comments, highlighting the corresponding revisions made in the updated version of the manuscript.

I . Reviewer 1

Conditional comments:

Comments 1: *The authors revised a paragraph of "In Fig. 9, ..." from line of 10 in the right part of page 8 in the manuscript. I think "the correlation distance of 1 chip" in line of 15 in the page should be "the correlation distance of 1/2 chips". If so, please modify it.*

Response 1: Thank you for your valuable comment. We appreciate your attention to detail. As per your suggestion, we have revised the paragraph in question. The term "the correlation distance of 1 chip" has been updated to "the correlation distance of 1/2 chips" on page 10 of the revised paper:

"In Fig. 9, there is a clear downward trend of multipath when combined with narrow correlation technology. In particular, the envelope peak value of the correlation distance of 1/8 chip can be further reduced by 4/5, and the effective length of the envelope can be reduced by 7/8 compared to the correlation distance of 1/2 chip. As a result, the envelope peak value can be reduced by 98%, equivalent to around 0.005 chips. Moreover, approximately 87.5% of the overall time delay of the multipath error can be minimized to approach zero when random errors in the pseudorange measurements are not considered, indicating the theoretical effectiveness of the antenna jitter."